# Asymmetric gating of a human hetero-pentameric glycine receptor

Xiaofen Liu [1] & Weiwei Wang [1] ✉

Hetero-pentameric Cys-loop receptors constitute a major type of neurotransmitter receptors that enable signal transmission and processing in the nervous system. Despite intense investigations into their working mechanism and pharmaceutical potentials, how neurotransmitters activate these receptors remains unclear due to the lack of high-resolution structural information in the activated open state. Here we report near-atomic resolution structures resolved in digitonin consistent with all principle functional states of the human α1β GlyR, which is a major Cys-loop receptor that mediates inhibitory neurotransmission in the central nervous system of adults. Glycine binding induces cooperative and symmetric structural rearrangements in the neurotransmitter-binding extracellular domain but asymmetrical pore dilation in the transmembrane domain. Symmetric response in the extracellular domain is consistent with electrophysiological data showing cooperative glycine activation and contribution from both α1 and β subunits. A set of functionally essential but differentially charged amino acid residues in the transmembrane domain of the α1 and β subunits explains asymmetric activation. These findings provide a foundation for understanding how the gating of the Cys-loop receptor family members diverges to accommodate specific physiological environments.

Mammalian Cys-loop receptor superfamily encompasses ionotropic neurotransmitter receptors, including cation-selective type 3 serotonin receptors (5-HT3) and nicotinic acetylcholine receptors (nAChR), as well as anion-selective glycine receptors (GlyR) and GABA(A) receptors. These receptors underly fast synaptic and extrasynaptic neurotransmission throughout the nervous systems. They also mediate communication between the nervous and locomotive systems[1]. A myriad of neurological disorders, including Alzheimer's disease, schizophrenia, epilepsy and autism[1–6], as well as locomotive problems such as hyperekplexia and myasthenic syndrome[7–9], are caused by or associated with their dysfunction.

Despite decades of intense research, (near) atomic resolution structures in the activated open state have only been characterized for the homomeric receptors, including the homomeric glycine receptors that are found almost solely in embryos[10–13], and the homomeric α7 nAChR and 5-HT3A receptors that represent only one of the many subunit compositions found in tissues[14,15]. All the homo-pentameric structures point to an open mechanism through symmetric expansion of the ion conduction pathway[10,11,16,17] (with the exception of 5-HT3A, where asymmetric dilation was recently reported[15]). For the predominant heteromeric receptors, although the variations in sequences and chemical properties in the constituting subunits may lead to asymmetric gating, the cooperativity of neurotransmitters in receptor activation suggests otherwise. Unfortunately, structures of heteromeric receptors have only been captured in either the resting/inhibited closed state or the desensitized state[18–20], leaving this conundrum unresolved and details of open state unclear.

GlyRs are major Cys-loop receptors that mediate fast inhibitory neurotransmission in the spinal cord and brain[13,21,22]. Dysfunction of GlyR causes the congenital disorder hyperekplexia[7,8,23–25]. It is a therapeutic target in neuropathic pain[8,26–28] and related to autism and other neurological disorders[4,13,22,29]. The widely studied stereotypical

[1]Department of Biophysics, University of Texas Southwestern Medical Center, Dallas, TX, USA. ✉e-mail: Weiwei.Wang@UTSouthwestern.edu

homomeric GlyRs contain only the α subunits and are rarely found in adult animals. On the contrary, the heteromeric GlyRs containing both the α and the β subunit are found throughout the central nervous system[13]. Structural information on heteromeric GlyRs has only become available very recently, revealing an unexpected 4:1 α:β subunit stoichiometry[18,30,31]. Unfortunately, none of the structures reported was in the activated open state, despite a "semi-open" state in one study[30] had its ion conduction pore asymmetrically expanded to a size just shy of the predicted range of open GlyR pores[22,30,32].

In this study, we resolve near-atomic resolution structures of the α1β GlyR with native function in digitonin with pore geometries consistent with all principle functional states throughout its gating cycle: apo (closed), glycine-bound desensitized and glycine-bound open. In combination with electrophysiology and mutagenesis experiments, we show that glycine binding to the orthosteric site results in highly cooperative and pseudo-symmetric conformational changes in the extracellular domains (ECD), explaining the cooperativity of glycine in the activation of GlyRs. However, due to the distinct characteristics of the α and β subunit transmembrane domains (TMD), symmetric activation in ECD resulted in asymmetric structural rearrangements in the TMD, leading to asymmetric pore dilation that is distinct from homomeric GlyRs. Such differential responses in ECD and TMD reconciliate the cooperativity in glycine activation and differences in subunit properties in an asymmetric gating mechanism. This mechanism explains how mutations of a charged amino acid residue pointing away from the ion conduction pore could diminish heteromeric GlyR conductance and cause hyperekplexia.

## Results

### Overall architecture of the human α1β GlyR

To understand the gating mechanisms of the human α1β GlyR without complications from function-modifying antibodies and limitations in compositional homogeneity with purification from native tissues, we generated constructs of the α1 and β subunits (α1em and βem, respectively) that form α1emβem GlyR and exhibit indistinguishable

function as wild type. α1em contained a partial truncation of the unstructured M3-M4 loop (Supplementary Fig. 1a). βem has been described previously[33], which contained substitution of GFP for part of the unstructured M3-M4 loop that does not bind with synaptic scaffolds (Supplementary Fig. 1b). These constructs greatly improved biochemical properties of heteromeric GlyR (Supplementary Fig. 1c) during purification. In electrophysiological recordings, α1emβem GlyR and wild-type α1wtβwt GlyR shared the same apparent glycine $EC_{50}$ ~ 100 μM, Hill coefficient ($n$ ~ 1.7) and similar current magnitude across cells (Fig. 1a, b; Supplementary Fig. 1d, e). Apparently, α1emβem GlyR well recapitulates the functional properties of wild-type GlyR.

Through single-particle cryo-EM analysis, we resolved structures of α1emβem GlyR both in the absence and in the presence of glycine. The density arising from GFP fusion on the βem subunit served as a fiducial marker to differentiate the β subunit from the structurally similar α subunits[30] (Fig. 1c; Fig. 2a, b, i, j), without the use of available subunit-specific antibodies that alter function[18]. The overall resolution ranged from 3.6 to 4.1 Å, with local resolutions extending well beyond 3.0 Å for many regions of interest (Supplementary Fig. 2d–f, l–n). These density maps allowed for unambiguous model building for most part of the protein (Supplementary Fig. 3; Supplementary Table 1). In all the structures determined, either with or without glycine, 4 α1 subunits and 1 β subunit were identified (Fig. 1c, d), consistent with the recent discovery of 4:1 α2:β stoichiometry of human α2β GlyR and heteromeric GlyR purified from porcine tissue[18,30]. These structures suggest that 4:1 α1:β stoichiometry is an intrinsic property of α1β GlyR, not dependent on the presence of other GlyR subunits (such as α2) or unidentified factors/chaperons in native tissues[18].

### α1β GlyR in the closed and glycine-bound states

In the absence of glycine (apo state), a single conformation was resolved that corresponds to the closed states (Fig. 2a–c). All the orthosteric sites are empty (Fig. 2a). The activation gate[11] at 9' is tightly constricted, with the 9' Leu sidechains pointing toward the pore (Fig. 2b, c). The minimal pore radius is -1.8 Å, too narrow for Cl⁻ to pass (Fig. 2m black). This conformation resembles that of homomeric GlyR in the closed state[11], as well as the pseudo-symmetrical α2β-strychnine complex in the closed state[30].

In the presence of glycine, two structures were resolved with widened pores sufficient for Cl⁻ to pass through (Fig. 2d–i). In one structure (gly-1), clear glycine densities are found in all five orthosteric sites (Fig. 2d). Compared to the apo state, 9' Leu side chain flipped away from the pore, combined with outward movement of the M2 helix, resulted in an open activation gate[11]. The −2' position dilated in an apparently asymmetrical manner (Fig. 2 e, f). The minimal radius along the pore is -3.1 Å (Fig. 2m yellow), sufficient to allow partially hydrated Cl⁻ through and within expected range of physiological pore sizes of open GlyR[12,32]. Considering that minimal pore radii for all reported open GlyRs (homomeric) is 3.0 ± 0.17 Å at the level of −2' (Supplementary Table 2), this structure is most consistent with α1β being in its open state. Comparison of this open α1β GlyR pore with that of homomeric α1 GlyR (PDB ID: 6UD3) in the open state supported by molecular dynamics simulations (Supplementary Fig. 6k, l) reveals similar pore geometry, further supporting this annotation. In the other structure (gly-2), the orthosteric sites at all 3 α/α interfaces have clear glycine densities, while sites at α(+)/β(−) and β(+)/α(−) interfaces are empty (Fig. 2g, Supplementary Fig. 3). Nonetheless, the pore also dilated in an asymmetrical manner, but to a larger extent, resulting in a minimal radius of 3.8 Å (Fig. 2h, i, m cyan), which is too large for a physiologically open GlyR. Over-widened pore is not likely a result of truncation in the unstructured M3-M4 loops considering the functional equivalence of α1emβem GlyR with wild type (Fig. 1; Supplementary Fig. 1). In addition, a similar "expanded open" conformation has been recently reported of full-length homomeric GlyR in SMA nanodiscs[12], suggesting that it is not specifically resulting from the

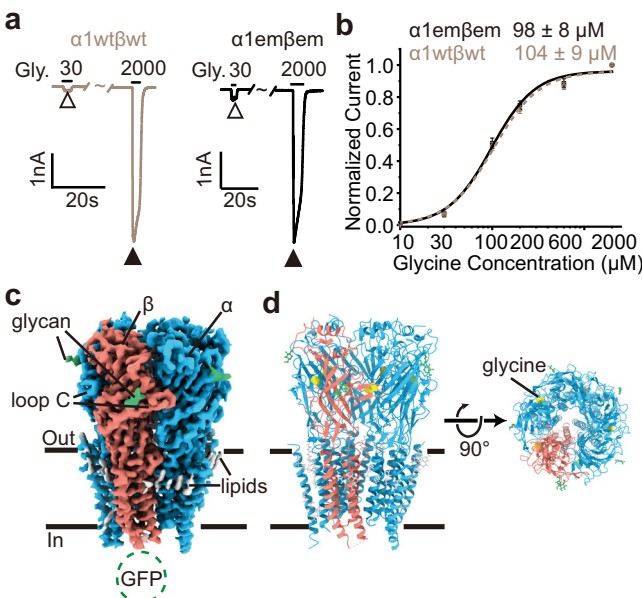

**Fig. 1 | Dose response of glycine and overall structure of the α1β GlyR. a** Typical glycine response of α1wtβwt and α1emβem at 30 μM and 2 mM. **b** Dose response with Hill fits (lines). Data are represented as mean ± SEM (*n* = 7 cells). Source data are provided as a Source Data file. **c** Side view of cryo-EM map of α1β GlyR−glycine complex. α subunits, β subunit, N-glycan, lipids, and glycine are respectively colored in sky blue, salmon, green, gray, and yellow. **d** Side (left) and top-down (right) view of the atomic models.

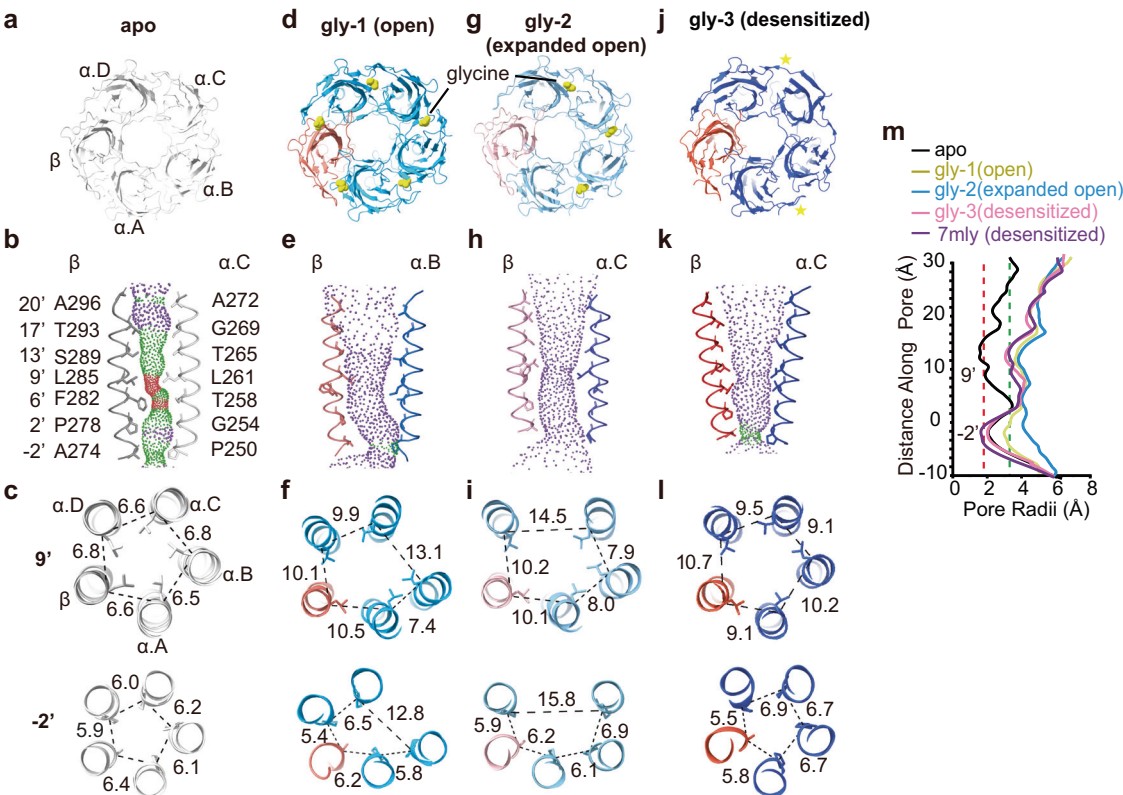

**Fig. 2 | α1β GlyR structures in major functional states. a, d, g, j** Top-down views in the apo (**a**), gly-1(open) (**d**), gly-2(expanded open) (**g**) and gly-3(desensitized) (**j**) states. α1 subunits and β subunits are colored in different shades of blue and orange, as indicated, except for in the apo state where β is gray and α1 is white. The yellow spheres represent glycine. Yellow stars indicate pockets with markedly glycine density. **b, e, h, k** Ion permeation pathways in the apo (**b**), gly-1(open) (**e**), gly-2(expanded open) (**h**), and gly-3(desensitized) (**k**) states. M2 helices are shown as cartoon and the sidechains of pore-lining residues as sticks. Purple, green, and red spheres define radii of >3.3 Å, 1.8–3.3 Å, and <1.8 Å, respectively. **c, f, i, l** Cross-sections of M2 helices at residues 9' (top) and −2' (bottom) in the apo (**c**), gly-1(open) (**f**), gly-2(expanded open) (**i**) and gly-3(desensitized) (**l**) states, with distances between neighboring Cα shown in Å. **m** Plot of pore radii calculated by the HOLE program for the apo (black), gly-1(open) (yellow), gly-2(expanded open) (sky blue), gly-3(desensitized) (hot pink) states, as well as that of porcine desensitized state (PDB ID: 7mly, purple). Source data are provided as a Source Data file.

digitonin detergent that was used. Indeed, a comparison of currently available "expanded open" state pore geometries (Supplementary Table 2, Supplementary Fig. 6m) shows that they are largely similar. A third structure (gly-3) was resolved in the α1emβem GlyR–glycine complex that corresponds to the desensitized state. Two out of five orthosteric sites had densities of glycine (Fig. 2j; Supplementary Fig. 3). However, due to the lower overall resolution (~4.1Å), we did not explicitly model glycine, instead only indicated these two positions in Fig. 2j. The activation gate around 9' Leu is open. The desensitization gate is constricted to a radius of ~2 Å (Fig. 2k, l, m pink), rendering the channel non-conductive. The pore shape in this structure is most similar to that of α2β GlyR[30], and GlyR purified from porcine tissue in the desensitized state[18] (Fig. 2m pink and purple), retaining mostly 5-fold pseudo-symmetric shape in the TMD. Intriguingly, non-protein densities have been observed in the glycine-bound states, which might complicate the assignment of functional state and hint at lipid-dependent gating (Supplementary Fig. 7, discussed later).

The above structures, consistent with properties in all the major functional states, depict a gating mechanism that has a resemblance but is also very different from homomeric GlyRs. Homomeric GlyRs maintain the 5-fold symmetric structure throughout the gating cycle, with all five orthosteric pockets in an identical ligand-binding state[10–12,17,34,35]. α1β GlyR changes its symmetry during gating and exhibit varying glycine occupancy in the five orthosteric sites. This provides a unique opportunity to understand how asymmetrical gating works in a multimeric ligand-gated ion channel containing non-equivalent subunits.

## Symmetric glycine-induced response in the ECD

Glycine binding induced similar conformational rearrangements across all five subunits. The binding of agonists is known to induce a compact conformation in the orthosteric site[10,11,18,36,37]. In our apo structure (Fig. 2a), all five binding pockets are empty and in the same apo conformation (Supplementary Fig. 4a). In the gly-1(open state) structure, glycine is found in all five pockets, resulting in the same compact conformation across all pockets (Fig. 3a, Supplementary Fig. 4b). However, in the gly-2 (expanded open) and gly-3(desensitized) states, clear glycine densities are observed only in a subset of orthosteric pockets (3 and 2 out of 5, respectively) (Fig. 2g, i). Partial glycine occupancy is unlikely a result of limited resolution as clear differences in densities were observed between bound and free sites with similar local resolutions (Supplementary Fig. 3c, d). Nonetheless, the same compact conformation was observed across all five pockets, irrespective of whether glycine is bound (Fig. 3b, c). In addition, ECDs from all five subunits showed similar rocking motions that propagate to TMD during channel activation (Fig. 3h). Such symmetric change in the ECD upon glycine binding more likely suggests cooperativity among orthosteric pockets, which is indicated by Hill slopes ranging from ~2 to 5 in our and previously reported glycine-dose response curves (Fig. 1 and Supplementary Fig. 5)[22,30,38,39]. The following electrophysiological experiments further demonstrate that α1β GlyR gating does not require fully occupied allosteric sites.

All orthosteric pockets apparently contribute similarly to glycine activation of α1β GlyR. This is indicated by the same conformation across all binding pockets irrespective of subunit type. To further test

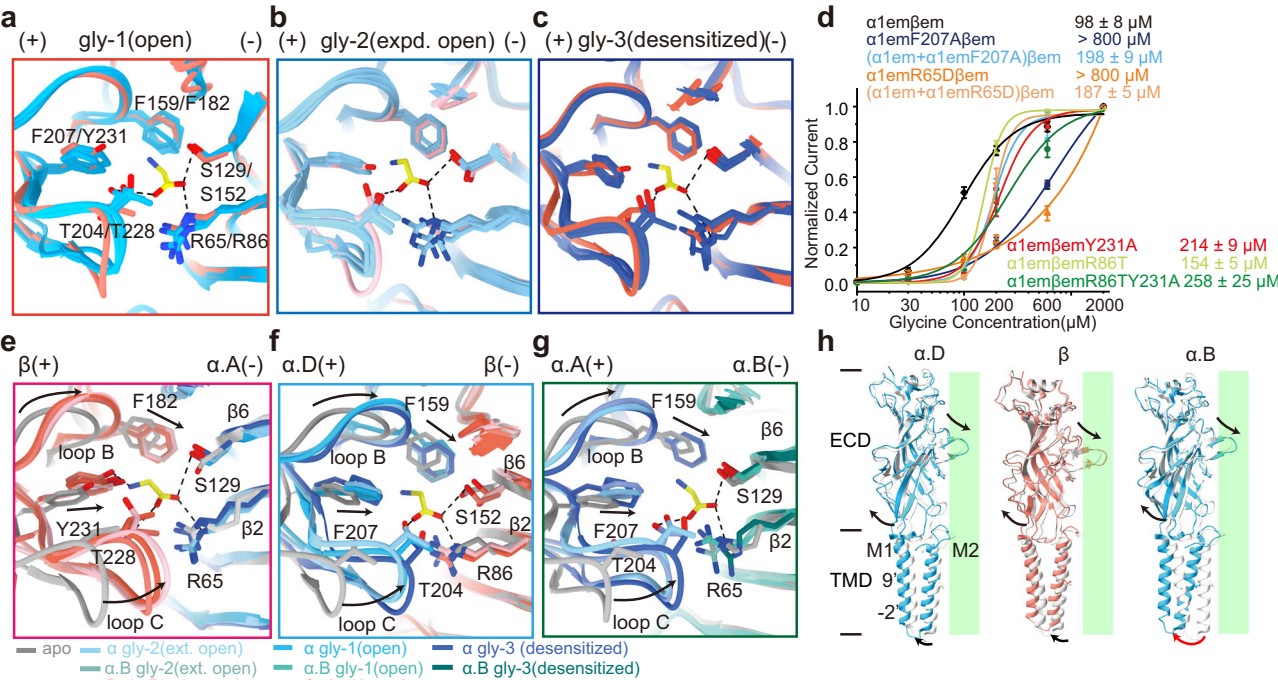

**Fig. 3 | Propagation of glycine-induced structural changes. a–c** Overlay of all 5 glycine binding pockets in each of the gly-1(open) (**a**), gly-2 (expanded open) (**b**) and gly-3(desensitized) (**c**) states, with (+) and (−) sides indicated. The key amino acids for glycine binding are shown. **d** Dose response of glycine for α (+) (α1emF207Aβem), α(+) (α1emR65Dβem), β(+) side (α1emβemY231A), β(−) side (α1emβemR86T) and β double sides (α1emβemR86TY231A) mutants. Data are represented as mean ± SEM ($n = 9$ cells for α1emF207Aβem, $n = 8$ cells for (α1em+ α1em)F207Aβem, $n = 9$ cells for α1emR65Dβem, $n = 7$ cells for (α1em + α1emR65D) βem, $n = 6$ cells for α1emβemY231A, $n = 7$ cells α1emβemR86T, for $n = 8$ cells for α1emβemR86TY231A). $EC_{50}$ from Hill fits are listed. Source data are provided as a Source Data file. **e–g** The orthosteric pocket changes from apo state to gly-1(open), gly-2(expanded open) and gly-3(desensitized) states at α.D(+)β(−) (**e**), β(+)α.A(−) (**f**) and α.A(+)α.B(−) (**g**) subunit interfaces. The key amino acids are indicated. **h** Superimposition of individual subunits in the apo (α: white, β: gray) and gly-1(open) states (α: blue, β: salmon). The rotational motion of ECD propagates to TMD in different ways. Arrows indicate the direction of motion. Green rectangles represent the pore axis.

this, we mutated amino acid residues (α:F207 or β:Y231, etc., see Fig. 3a) that is important for glycine binding and performed glycine titration (Fig. 3d). Mutating α:F207 in α(+) side or α:R65 in α(−) side resulted in dramatic decrease in apparent glycine affinity, to the point where only a lower limit of $EC_{50} \sim 0.8$ mM can be estimated (Fig. 3d, blue, orange), while mutating the homologous β:Y231 in β(+) side or β:R86 in β(−) side only resulted in ~2-fold increase in $EC_{50}$ (Fig. 3d red, yellow), with similar maximum current (fully activated, Supplementary Fig. 5). This is because there is only 1 β subunit but 4 α subunits in each GlyR. Consistent with this, mutating a subset of α:F207 or α:R65 resulted in a similar effect as the β:Y231 or β:R86 mutation (Fig. 3d, light blue, light orange). This phenomenon is consistent across extensive mutagenesis experiments in glycine binding pockets (Supplementary Fig. 5). The equivalence in response to glycine binding of all subunits is expected considering the cooperativity among these sites, which ensured a pseudo-5-fold symmetric structure in the extracellular domain throughout the gating cycle (Supplementary Fig. 6a, c, e, g). In addition, mutating both the (+) and (−) sides of β subunit, leaving only three intact sites at α/α interfaces, only increased $EC_{50}$ by ~3 folds (Fig. 3d, green) and did not affect maximum current (Supplementary Fig. 5). These observations also suggest that activation of α1β GlyR does not require binding to all five allosteric pockets, consistent the partially occupied pockets in the activated glycine-bound structures.

Interestingly, glycine-induced structural changes in the ECD are similar across all glycine-bound states(Fig. 3e–g). Orthosteric pockets at the β(+)α(−) (Fig. 2e), α(−)β(+) (Fig. 2f), and α(+)α(−) (Fig. 2g) sub-unit interfaces showed similar contraction upon glycine binding, which in turn induced a rotational motion of the ECD (Fig. 3h)[10,11]. ECD rotation moved the directly connected extracellular end of the M1

helix away from the conduction pore, resulting in an outward and symmetric expansion at the ECD-TMD interface (Supplementary Fig. 6i, j). Fivefold pseudosymmetry is maintained through all functional states in ECD and the extracellular end of TMD. However, the conformations of TMD near the intracellular side become more distinctive between different subunits (Fig. 3h). Apparently, the TMD of heteromeric GlyR converts the same symmetric input from ECD into different, sometimes structurally asymmetrical, functional states.

## Asymmetrical gating in the α1β GlyR TMD

α1β GlyR in the apo and gly-3(desensitized) states, but not in the gly-1/ 2(open/expanded-open) states, retained 5-fold pseudosymmetry, despite that the ECD remains largely symmetric throughout the gating cycle (Fig. 4a, d; see also Fig. 2 and Supplement Fig. 6). In the gly-1(open) state, the extracellular end of all five TMDs moved radially away from the conduction pore, resulting in a symmetric expansion similar to the gly-3(desensitized) state (the pore-lining M2 helices are shown in Fig. 4a, b, d top panel). However, the expansion became asymmetrical near the intracellular end−in addition to small radial expansion, one of the α1 subunits (chain B) moved away in both radial and tangential directions, creating a wider spacing from one adjacent α subunit (chain C) (Fig. 4b lower panel). This movement resulted in an asymmetrically widened desensitization gate that allows the conduction of Cl⁻. The gly-2(expanded open) state showed resembling but more extended structural rearrangements near the desensitization gate−two of the α1 subunits (chain B and C) moved in both radial and tangential directions, creating an asymmetrical wide-open pore. Intriguingly, none of the α1 subunits adjacent to the β subunit experienced such large movements, which we believe are resulting from amino acid residues unique to each subunit type, as discussed below.

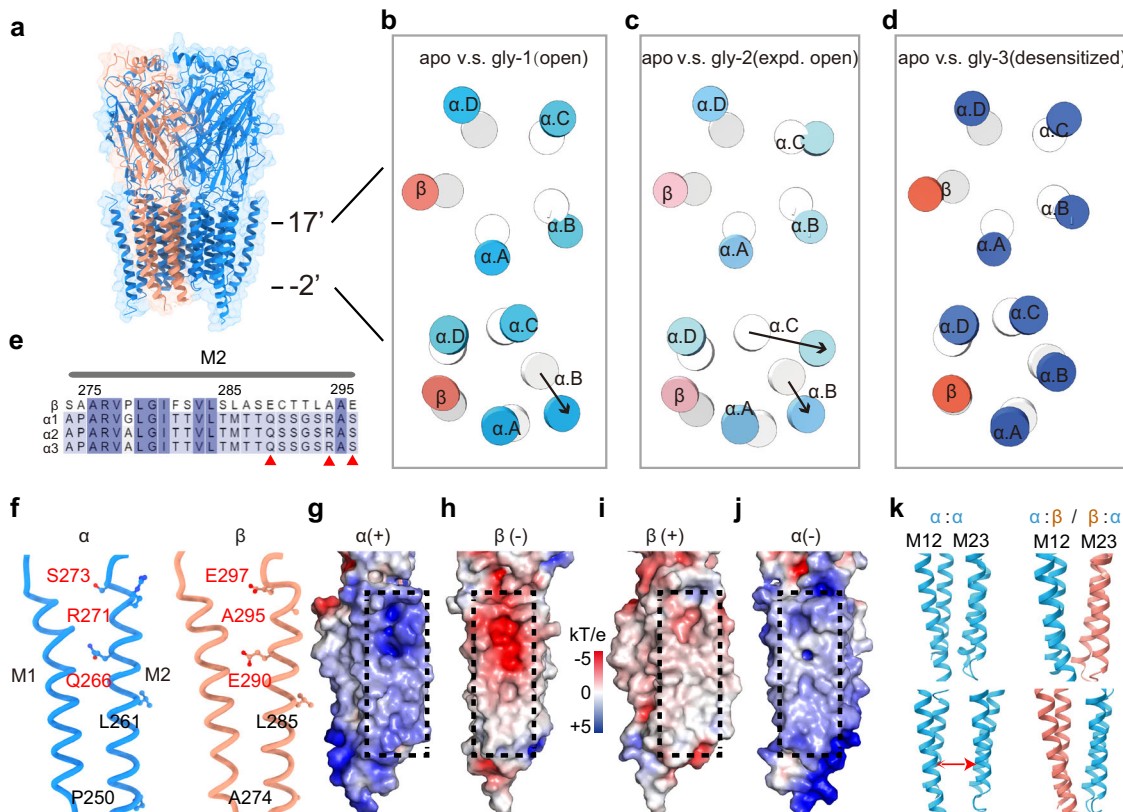

**Fig. 4 | Asymmetric conformational changes in TMD. a** Side view of the α1β GlyR. β: dark salmon; α: blue. The position of 17' and −2' are labeled. **b–d** Cross-sections at 17' and −2' of M2 of the apo (gray/white) and gly-1(open)/gly-2(expanded open)/gly-3(desensitized) states. α1 subunits and β subunits are colored in different shades of blue and orange. Black arrows indicate the movement direction. **e** Sequence alignment of M2 helices among human GlyR β, GlyR α1, GlyR α2, GlyR α3. **f** The position of key amino acids in the helix. The corresponding amino acids are marked in panel. **g–j** TMD surface colored according to electrostatic potentials (calculated using APBS tools) at α (+) (**g**), β (−) (**h**), β(+) (**i**) and α(−) (**j**) interfaces. Dashed boxes indicate subunit interface regions shown in (**f**). **k** The cartoon representation of the TMD interfaces between α (blue)-β (salmon) and α-α subunits in the gly-1(open) state. The red arrow indicates the increased distance between adjacent α subunits.

Differentially charged amino acid residues in the α1 and the β M2 helices promote asymmetrical gating. The pore-lining M2 helix is one of the least conserved regions between the α and β subunits in amino acid sequences (Fig. 4e), harboring disease-causing mutagenetic sites and resulting in distinctive functional features including glycine dose-response, single-channel conductance and picrotoxin sensitivity in homo- and heteromeric GlyRs[13,22,30,38,40]. A set of unique amino acid residues resulted in opposite electrostatic fields at subunit interfaces in TMD of α1 and β subunits (Fig. 4f–j). The combination of two neutral and one positively charged residues, α1:S273/R271/Q266, makes the TMD of α1 positively charged at both the (+) (Fig. 4g) and (−) (Fig. 4j) sides. The β subunit has two negatively charged residues and one hydrophobic residue at corresponding positions; β:E297/A295/ E290, which renders negative potentials at both (+) and (−) sides (Fig. 4h, i). Opposite potentials promote the interaction between the α1 and β subunit TMDs (Fig. 4g binds with h, i binds with j), especially in the hydrophobic environment inside the membrane, leading to a stable α1-β-α1 assembly throughout the gating cycle (Fig. 4b–d, Fig. 2c–l). However, the same positive potential contributes to repulsion between α1-α1 interfaces, increasing the likelihood of widening in α1-α1 interfaces. This explains the widening of some α1-α1 but not the α1-β-α1 interfaces in both the open (Fig. 4k) and expanded-open states. In both cases, a mostly rotational motion (Fig. 5a black arrow) of one α1 TM created larger widening near the −2' gate than at α1: R271 since −2' is further away from the pivot point−the TM2-TM3 loop (Fig. 5a). To differentiate electrostatic repulsion from specific types of charge (positive) at α1:R271 in facilitating GlyR opening, we generated α1:R271A (neutral) and α1:R271E (negative) mutants and characterized

their activation by glycine. Clearly, the α1:R271A neutral mutation that eliminated repulsion between α subunits almost completely abolished the gating of GlyR, while the reversed charge mutation α1:R271E retained normal GlyR function (Fig. 5a, b). Widening of α1-α1 subunit distance close to −2' gate during asymmetric conformational change is functionally correlated with the opening of the α1β GlyR. We designed cysteine-pair mutations between two α1 subunits, α1:A251C+V253C, that are within disulfide bonding distance (Fig. 5a solid red line, ~4.2 Å between Cβs) before asymmetric widening but too far after (Fig. 5a dashed red line, >7 Å between Cβs). Application of hydrogen peroxide diminished α1β GlyR activity with these mutants, which can be reversed using DTT (Fig. 5c). Decrease of activity by $H_2O_2$ was not observed in control mutants (α1:A251C+A302C) that do not favor inter-subunit disulfide bonding (Fig. 5d). In addition, cross-linking the β subunit with its neighboring α subunits either at the α/β or β/α interfaces did not decrease activity (Fig. 5e, f). These experiments suggest that subunit widening at the α/α interfaces promotes the opening of α1β GlyR gating.

## Discussion

We have resolved the structures of human α1β GlyR consistent with all its major functional states, which depicts the structural rearrangements through the gating cycle (Fig. 6). In the apo state, 5-fold pseudosymmetry is maintained in the whole α1β GlyR, with tightly constricted ion conduction pore (Fig. 6a). When glycine binds, the ECD of all five subunits experienced similar rotational motion, maintaining pseudosymmetry. ECD rotation pulls on the extracellular end of TMD, resulting in the same radial motion away from the pore in all five

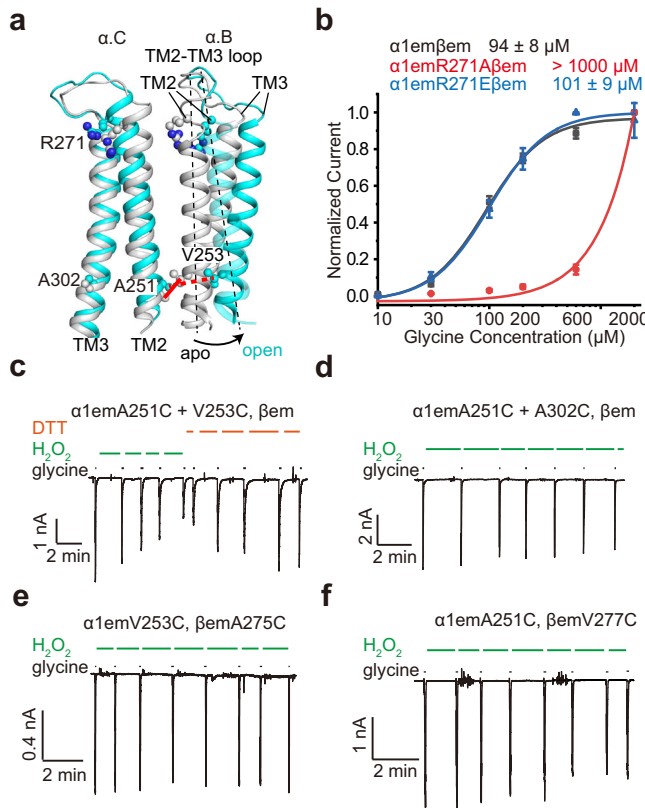

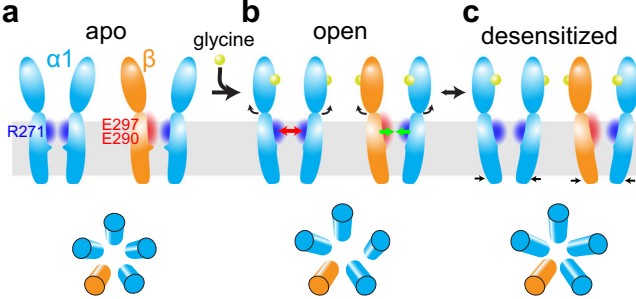

**Fig. 6 | Proposed gating mechanism of the α1β GlyR.** Adjacent α (blue) and β (orange) subunits are shown with respectively charged amino acid residues and surrounding electrostatics (positive: blue; negative: red). **a** In the apo state, heteromeric α1β GlyR is pseudo-symmetrical with a closed pore. **b** Upon glycine (yellow spheres) binding, conformational changes in ECD cause the widening of the extracellular end of the TMD. Electrostatic repulsion between adjacent α subunits makes them easier to separate (red arrow). On the contrary, the opposition electrostatics of α and β subunits ensure that they stay close (green arrow). This imbalance in electrostatic potentials in the TMD region promotes an asymmetric gating mechanism. **c** The sustained binding of glycine transitions α1β GlyR to a desensitized state, returning to a more pseudo-symmetrical conformation.

**Fig. 5 | Asymmetric widening correlates with GlyR α1β opening. a** Rotational motion pivoting around the M2-M3 loop of adjacent α1 subunits from apo (gray) to gly-1(open) (cyan) states, with positions of positive charge at α1:R271 and cysteine mutations indicated. **b** Glycine dose response of α1emβem ($n = 7$ cells), α1emR271Aβem ($n = 5$ cells) and α1emR271Eβem ($n = 9$ cells) GlyR. Data (mean ± SEM) and EC$_{50}$ from Hill fits (lines) are listed. Source data are provided as a Source Data file. **c** Representative whole-cell recordings ($n = 5$ cells) of α1em(A251C +V253C) βem, H$_2$O$_2$: 50 mM, DTT: 5 m, glycine: 200 μM. **d** Representative whole-cell recordings ($n = 4$ cells) of α1em(A251C+A302C)βem. H$_2$O$_2$: 50 mM, glycine: 200 μM. **e** Representative whole-cell recordings ($n = 3$ cells) of α1emV253CβemA275C. H$_2$O$_2$: 50 mM, glycine: 200 μM. **f** Representative whole-cell recordings ($n = 5$ cells) of α1emA251CβemV277C. H$_2$O$_2$: 50 mM, glycine: 200 μM.

subunits (Fig. 6b). However, due to electrostatics repulsion, the pore-lining M2 helix of one α1 subunit moves in a rotating motion pivoting on M2-M3 loop, tangentially away from the adjacent α1 subunit near the intracellular end. Since the β subunit carries opposite electrostatic charges, it remains attracted to neighboring α1 subunits. In this way, the pore is dilated in an asymmetrical manner, representing one possible open state of α1β GlyR. When desensitization happens, the intracellular end of M2 helices of all subunits collapses back into a more pseudo-symmetrical conformation that is thermodynamically stable[10,11,22,41], stopping Cl⁻ conduction (Fig. 6c).

The ECD and TMD of α1 and the β subunits contribute differently to the assembling and gating of α1β GlyR. Although the unique ECD of the β subunit dictates the 4:1 α:β stoichiometry in heteromeric GlyRs[30,39,42], it contributes similarly to glycine EC$_{50}$ of α1β GlyR. All five orthosteric pockets experience similar conformational changes from the close to the open/desensitized states, regardless of subunit types. This leads to identical structural changes in the extracellular end of TMD for all the α1 and β subunits (Fig. 2). The specific patterns of glycine occupation in the five available sites presented in this work mostly likely only show a subset of all possible conformations and unrelated to specific TMD conformations. We also note that limitations in resolution, especially in the gly-3(desensitized) state, might complicate the precise assignment of occupied sites. The TMDs, on the

other hand, do not affect the assembly but instead determine functional properties of heteromeric GlyRs and contain multiple disease-causing mutagenesis sites[13,38,40]. Mutation of one residue to remove the positive charge, R271L/Q/P, of the α1, is known to diminish Cl⁻ conduction and cause hyperekplexia in a dominant manner through unclear mechanism[7,23,25]. Recently it was shown that R271 (19′) allows conduction not through locally concentrating Cl⁻, but by creating electrostatic repulsion between subunits to allow the opening of the pore[43]. This is consistent with our structural observation of widened α1-α1, but not α1-β subunit distances in the open state because α1 and β subunits are oppositely charged at equivalent positions (Figs. 4 and 5). We believe asymmetrical expansion of pore is one likely mechanism for a heteromeric GlyR to open. Considering the multiple single-channel conductance states reported of heteromeric GlyRs[38,44], it is tempting to speculate that multiple open state conformations exist, and some remain to be identified.

Non-protein densities near or in the conduction pathway may hint at a more complicated gating mechanism of human heteromeric GlyRs. When evaluating the functional state each structure represents, it is common to investigate pore geometry and chemical properties based on protein atomic model[11,18,32,45,46]. However, non-protein densities are quite often found near or in the conduction pathway with unclear identities and are thus usually not modeled. This is indeed the case for all available mammalian heteromeric GlyR structures in the glycine-bound states both in detergents and in nanodisc[18,30], including the ones presented here. Particular interesting is those in the gly-1(open) and gly-2(expanded open) states since they may play structural, as well as gating roles (Supplementary Fig. 7). The widening of the α-α interface in the gly-1(open) state is limited, with weak and scattered densities at the interface (Supplementary Fig. 7a, b, red mesh), indicating weak/transient occupancy. Relatively weak tubular densities are present in the pore (Supplementary Fig. 7a, b, yellow mesh), resembling a beads-on-a-string shape that breaks up into blobs at higher thresholds, reminiscent of less-resolved ionic densities. In contrast, in the gly-2(expanded open) state (Supplementary Fig. 7c, d), strong and connected densities are found in the widened α-α interface, with stronger and more inter-connected lipid-like densities in the pore. These densities suggest a more stable occupancy, likely structurally supporting this conformation with wider α-α interface distance. In the case that densities in the pore represent hydrophobic plugs, they would render the channel non-conductive in this glycine-bound state.

Although lipid-involved gating of ion channel pores is an established mechanism[47,48], more investigation is clearly required to evaluate its relevance in α1β GlyR function.

Digitonin provided the best stability in biochemical purification and structure determination for α1β GlyR (Supplementary Fig. 1f–i), likely due to its ability to preserve native structure/function of multiple classes of membrane proteins[49] (likely through retaining native lipids, which were abundantly observed in density maps). Similar "expanded open" structures (Fig. 2g–i) have been reported in DDM[10] and saposin nanodiscs[12], whereas detergent (DDM/CHS) has also yielded heteromeric GlyR structure that is consistent with functional expectations[18]. Clearly, to evaluate the physiological relevance of all observed conformations, better membrane mimetics such as lipid vesicles[50], combined with deliberately selected lipid mixtures, holds the future for in vitro structure determination in a more native environment.

The β subunit being tightly tethered to its neighboring α1 subunits throughout the gating cycle is an interesting observation that is conceptually compatible with its role in the cellular organization of GlyRs. Post-synaptic scaffolding protein gephyrin binds to the intracellular M3-M4 loop of the β subunit and clusters GlyRs at post-synaptic membranes[51–53]. It is tempting to speculate that tight β-α1 interaction avoids aberrant channel activity due to force originating from anchoring. Since many Cys-loop receptors are clustered in post-synaptic densities, mechanisms that avoid excessive crossover between higher-order assembly and neurotransmitter activation would be desirable. Clearly, more evidence is required to evaluate this hypothesis.

## Methods

### Plasmid constructs

The human glycine receptor α1 (NCBI: NP_001139512.1) and β (NCBI: NP_000815.1) sequences were amplified from cDNA clones (McDermott Center, UT Southwestern Medical Center). The α1em sequence was derived by substitution of M3/M4 loop (residues R316-P381) s by GSSG peptide. For the βem construct, we used the previously described βem construct[30]. Region encoding M3/M4 loop (residues N334-N377) was replaced by GGSSAAA-monomeric enhanced green fluorescent protein-SGSGSG. A PA-tag (GVAMPGAEDDVV) and PreScission Protease site were inserted after the signal peptide. The α1em and α1 wild-type sequence was subcloned into a BacMam expression vector[54]. The β wild-type sequence was introduced into pLVX-IRES-ZsGreen1 vector (Clonetech) for electrophysiology. All α1em and βem mutants were generated using site-directed mutagenesis.

### Protein expression

The α1em and βem constructs were transformed into DH10BacY competent cells (Geneva Biotech) to produce bacmids. The bacmids were transfected into Sf9 cells (ATCC, CRL-1711) to generate baculovirus. Recombinant baculovirus titer was determined as described before[30,54]. Virus was added at a multiplicity of infection (MOI) of 2 (at 3βem:1α1em ratio) to HEK293S GnTI⁻ cells (ATCC, CRL-3022) cultures at a density of $2.5 \times 10^6$ cells/ml. To increase the expression level, 10 mM sodium butyrate was added, and the culture temperature was changed to 30 °C after transduction 12 h. Cells were collected after induction 60 h by centrifugation at 30,000 × g for 20 min at 4 °C and stored at −80 °C until further use.

### Protein purification and saposin nanodisc reconstitution trial

Cell pellets were thawed and resuspended in a by lysis buffer (40 mM Tris pH 8.0, 50 mM NaCl, 2 mM MgCl₂, 1 mM CaCl₂, 20 μg/ml Dnase, 2 μg/ml leupeptin, 2 μM pepstatin, 0.8 μM aprotinin, 0.2 mM PMSF) rotated at 4 °C for 30 min under constant stirring, followed by centrifugation at 40,000 × g for 20 min to collect cell debris. The cell debris was dounced and centrifugated at 40,000 × g at 4 °C for 20 min. The pellets were further homogenized and solubilized with buffer A

(40 mM Tris pH 8.0, 200 mM NaCl, 2 mM MgCl₂, 1 mM CaCl₂, 20 μg/ml Dnase, 2 μg/ml leupeptin, 2 μM pepstatin, 0.8 μM aprotinin, 0.2 mM PMSF, 0.75%(w/v) DDM, 0.075%(w/v) CHS and 0.075% (w/v) Na Cholate) for 40 min at 4 °C. Solubilized membranes were cleared by centrifugation at 40,000 × g for 30 min. Afterward, the supernatant was added to PA-tag antibody (NZ-1)[55] resin at RT. The resin was collected by a gravity column and washed with 10CV buffer B (20 mM Tris pH 8.0, 200 mM NaCl, 2 mM MgCl₂, 1 mM CaCl₂, 0.2 mM PMSF, 0.05% (w/v) DDM (Anatrace), 0.005% (w/v) CHS (Anatrace), 0.001% (w/v) Na Cholate (Anatrace)). Then, beads were mixed with PreScission protease (1:30 v/v) to cleave PA tag at RT for 1 h. The flow-through was collected, and resin was washed with 2CV buffer B. All proteins were pooled and concentrated to load onto Superose6 increase 10/300 GL column (GE Healthcare) in SEC buffer (20 mM Tris pH 8.0, 200 mM NaCl, 0.05%(w/v) DDM, 0.005% CHS).

Reconstitution of GlyR α1β into saposin nanodisc was modified from the published protocol. After small screening, a 1:30:200 molar ratio of α1β:saposin:brain polar lipids extract (BPE) (Avanti) was used. Mix protein and BPE at room temperature (RT) for 10 min. Saposin was added and the mixture was put at RT for another 2 min. Then dilution buffer (20 mM Tris pH 8.0, 200 mM NaCl) was added to the mixture and incubated on ice for 30 min. Bio-beads SM-2 (Bio-Rad) were added to the mixture and rotated overnight at 4 °C. Next morning, fresh bio-beads were added after the old bio-beads were removed for another 10 h. The mixture was centrifuged before loading onto Superose6 increase size exclusion column in SEC buffer (20 mM Tris pH 8.0, 200 mM NaCl).

### Protein purification for Cryo-EM data collection

Cell lysis and protein solubilization by detergent follow the same protocol as the protein purification of saposin nanodisc reconstitution. Briefly, solubilized membranes were cleared by centrifugation at 40,000 × g for 30 min. The supernatant was collected and added to PA-tag antibody (NZ-1)[55] resin at RT. The resin was collected and washed with 5CV buffer B and 5CV buffer C (20 mM Tris pH 8.0, 200 mM NaCl, 2 mM MgCl₂, 1 mM CaCl₂, 0.06% (w/v) digitonin (Sigma-Aldrich)). Then, beads were mixed with PreScission protease (1:30 v/v) to cleave PA tag at RT for 1 h. The flow-through was collected, and resin was washed with 2CV buffer C. All proteins were pooled and concentrated to load onto Superose6 increase 10/300 GL column (GE Healthcare) in SEC buffer (20 mM Tris pH 8.0, 200 mM NaCl, 0.06%(w/v) digitonin). Peak fractions were collected and concentrated to 6 mg/ml for grids freeze. For the sample with glycine, the buffer used throughout the purification process contained 2 mM glycine.

### Cryo-EM sample preparation, data collection and image processing

For apo α1emβem GlyR, the sample was vitrified without any ligand. For glycine-bound α1β GlyR, the sample was incubated for 1 h with 2 mM glycine on ice. Then, 1 × CMC final concentration (~3 mM) of Fluorinated fos-choline 8 (Anatrace) was added into the sample immediately before freezing. Grids (Quantifoil R1.2/1.3 400-mesh Au holey carbon grid) were glow-discharged. An FEI Vitrobot Mark IV Vitrobot (Thermo Fisher) was employed to plunge freeze the grids after application of 3 μl sample at 4 °C under 100% humidity.

Micrographs were collected using a Titan Krios microscope (Thermo Fisher) with a K3 Summit direct electron detector (Gatan) operating at 300 kV using the SerialEM data acquisition software. The GIF-Quantum energy filter was set to a slit width of 20 eV. Images were recorded in the super-resolution counting mode with a pixel size of 0.415 Å. Micrographs were dose-fractioned into 50 frames with a dose rate of 1.4 e⁻/Å/frame.

Twofold binning (0.83 Å pixel size after binning), motion correction and dose weighting of the movie frames were performed using the Motioncorr2 program[56]. CTF correction was carried out using the

CTFFIND 4 program[57]. The following image processing steps were carried out in RELION 3[58], as illustrated in Supplementary Fig. 2. Particles were initially picked using the Laplacian-of-Gaussian blobs and subjected to 2D classification to obtain good class-averages, which was then used as template for reference-based autopicking. The resulting particles were extracted with 4-fold binning for a further round of 2D classification (Supplementary Fig. 2a, i). Good 2D classes were selected and subjected to 3D classification using an initial model downloaded from the EMDB database (EMD-23148)[30]. For both the apo and glycine-bound samples, 1 out of 6 classes in 3D classification appeared with good density for the entire channel (Supplementary Fig. 2b, j). A single-density blob for GFP was identified for both the apo and glycine-bound samples. A further 3D classification into 4 classes with non-binned particles (0.83 Å pixel size) without particle alignment was performed. For the apo sample, partial signal subtraction[59] was performed to focus on the TMD. Two indistinguishable good classes were pooled, which resulted in a final of 29, 850 particles (Supplementary Fig. 2c). After reverting particles to un-subtracted version, CTF refinement, Bayesian polishing in RELION and non-uniform refinement[60] in cryoSPARC[61], an overall resolution of 3.6 Å was achieved, with local resolutions exceeding 3.0 Å in many regions (Supplementary Fig. 2d, f). For the glycine-bound sample, second 3D classification was performed using a mask excluding GFP and micelle, resulting in three good classes with distinct conformations (Supplementary Fig. 2k). After CTF refinement, Bayesian polishing in RELION and non-uniform refinement in cryoSPARC, overall resolutions of 3.6 Å (21, 676 particles), 3.9 Å (24, 487 particles) and 4.1 Å (30, 723 particles) were achieved for the gly-1 (open), gly-2 (expanded-open) and gly-3 (desensitized) states, with local resolutions exceeding 3.0 Å in many regions (Supplementary Fig. 2l, n). Resolutions were estimated by applying a soft mask around the protein densities with the Fourier Shell Correlation (FCS) 0.143 criterion. Local resolutions were calculated using Resmap[62].

## Model building and refinement

Models of GlyR α1β heteromer were built by fitting the structure of *Rattus norvegicus* α1β homomer glycine-bound state (PDB ID: 7MLY)[18] into the Cryo-EM density maps of GlyR α1β heteromer using Chimera[63] and Coot[64]. The atomic model was manually adjusted in Coot. The final models were refined with a real-space refinement module and validated with a comprehensive validation module in PHENIX package[65,66]. Fourier shell correlation (FSC) curves were calculated between the refined atomic model and the work/free half maps as well as the full map to assess the correlation between the model and density map (Supplementary Fig. 2e, m). Statistics of cryo-EM data processing and model refinement are listed in Supplementary Table S1. Pore radii were calculated using the HOLE program[46]. Figures were prepared in UCSF Chimera[63], ChimeraX[67], and PyMOL[68].

The final model of apo α1β GlyR contained the α1 and β subunit amino acids except the following: α1 subunit of chain A and chain C (total 367aa, 345aa built, 22aa not built) A1-P7,L314-L315, GSSG linker, E382 and V421- Q429; α1 subunit of chain B (total 367aa, 342aa built, 25aa not built) A1-P7, H311-L315, GSSG linker, E382 and V421- Q429; α1 subunit of chain D (total 367aa, 342aa built, 25aa not built) A1-M8, K312-L315, GSSG linker, E382-E383 and V421- Q429; β subunit (total 444aa, 338aa built, 106aa not built) K1-N32, GSSAAA-EGFP-SGSGSG insertion and V378-P442.

The final model of gly-2 (expanded open) α1β GlyR contained amino acids except the following: α1 subunit of chain A (total 367aa, 341aa built, 26aa not built) A1-P7, K312-L315, GSSG linker, E382-E383 and V421-Q429; α1 subunit of chain B (total 367aa, 339aa built, 28aa not built) A1-P7, R309-L315, GSSG linker, E382-E383 and R422- Q429; α1 subunit of chain C (total 367aa, 337aa built, 30aa not built) A1-M8, H311-L315, GSSG linker, E382-K385 and V421- Q429; α1 subunit of chain D (total 367aa, 340aa built, 27aa not built) A1-P7, H311-L315, GSSG

linker, E382-E383 and V421-Q429; The model of β subunit for gly-2 (expanded-open) is the same as apo.

The final model of gly-1 (open) α1β GlyR contained amino acids except the following: α1 subunit of chain A (total 367aa, 341aa built, 26aa not built) A1-P7, K312-L315, GSSG linker, E382-E383 and V421-Q429; α1 subunit of chain B (total 367aa, 340aa built, 27aa not built) A1-P7, R309-L315, GSSG linker, E382-R385 and V421- Q429; α1 subunit of chain C (total 367aa, 336aa built, 31aa not built) A1-M8, H311-L315, GSSG linker, E382-K386 and V421- Q429; α1 subunit of chain D (total 367aa, 340aa built, 27aa not built) A1-P7, H311-L315, GSSG linker, E382-E383 and V421- Q429; The model of β subunit for gly-1 (open) is the same as apo.

The final model of gly-3 (desensitized) state contained the α1 and β subunit amino acids except the following: α1 subunit of chain A (total 367aa, 335aa built, 32aa not built) A1-P7, Q310-L315, GSSG linker, E382-L387 and V421- Q429; α1 subunit of chain B (total 367aa, 336aa built, 29aa not built) A1-P7, Q310-L315, GSSG linker, E382-M384 and V421-Q429; α1 subunit of chain C (total 367aa, 337aa built, 30aa not built) A1-P7, H311-L315, GSSG linker, E382-K386 and V421-Q429; α1 subunit of chain D (total 367aa, 340aa built, 27aa not built) A1-P7, K312-L315, GSSG linker, E382-M384 and V421- Q429; The model of β subunit for gly-3 (desensitized) is the same as apo.

## Fluorescence-detection size-exclusion chromatography (FSEC) expression assay

In the FSEC assay, fluorescence was detected using the RF-20Axs fluorescence detector for HPLC (Shimadzu, Japan) (for EGFP, excitation: 480 nm, emission: 512 nm) as EGFP was fused into βem construct. Using 2 μl of Lipofectamine 3000 (Thermo Fisher Scientific, US), 1 μg of plasmid (at 1α1:3β ratio) was transfected into HEK293T cells for each well of 12 well plate. Cells were incubated in a $CO_2$ incubator (37 °C, 8% $CO_2$) for 48 h after transfection and solubilized with 50 μl buffer B for 1 h. After centrifugation (40,000 × $g$, 30 min), 50 μl of the sample was applied to a Superose6 Increase 10/300 GL column (GE Healthcare) equilibrated with buffer D (20 mM Tris pH 8.0, 200 mM NaCl,0.025% DDM) for the FSEC assay.

## Whole-cell patch clamp

The glycine $EC_{50}$ values were determined on α1β GlyRs expressed in HEK293T cells (ATCC, CRL-3216). Plasmids were transiently transfected using Lipofectamine 3000 reagent (Invitrogen). A total of 0.8 μg of DNA was transfected at 1α1:3β ratios for each 35 mm dish. Whole-cell recordings were made after 17–24 h transfected at room temperature. GFP fluorescence was used to identify the cells expressing the heteromeric α1β GlyRs. The bath solution contained (in mM): 10 HEPES pH 7.4, 10 KCl, 125 NaCl, 2 $MgCl_2$, 1 $CaCl_2$ and 10 glucoses. The pipette solution contained (in mM): 10 HEPES pH 7.4, 150 KCl, 5 NaCl, 2 $MgCl_2$, 1 $CaCl_2$ and 5 EGTA. The resistance of borosilicate glass pipettes is between 2–7 MΩ. For data acquisition, voltage held at −50 mV and a Digidata 1550B digitizer (Molecular Devices) was connected to an Axopatch 200B amplifier (Molecular Devices). Analog signals were filtered at 1 kHz and subsequently sampled at 20 kHz and stored on a computer running pClamp 10.5 software. Data analysis was performed by Origin 2018 software (Origin Lab). Hill1 equation was used to fit the dose-response data and derive the $EC_{50}$ ($k$) and Hill coefficient ($n$). For glycine activation, we used $I = I_0 + (I_{max} - I_0)\frac{x^n}{k^n + x^n}$, where $I$ is current, $I_O$ is the basal current (accounting mostly for leak, very close to 0), $I_{max}$ is the maximum current and $x$ is glycine concentration. All start point is fixed at 0 during fit. Measurements were from 4 to 11 cells, and mean and S.E.M. values were calculated for each data point.

## Reporting summary

Further information on research design is available in the Nature Portfolio Reporting Summary linked to this article.

## Data availability

The density maps for the cryo-em data have been deposited in the Electron Microscopy Data bank under accession codes EMDB-27553 (apo state), EMDB-27552 (gly-2, expanded open state), EMDB-27555 (gly-1, open state), MDB-27554 (gly-3, desensitized state). The coordinates have been deposited in the Protein Data Bank under accession codes 8DN3 (apo state), 8DN2 (gly-2, expanded open state), 8DN5 (gly-1, open state), 8DN4 (gly-3, desensitized state). The previously published structures 7MLY and 6UD3 used in our data analysis are also available from the PDB. Source data are provided with this paper.

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

## Acknowledgements

We thank Robbie Boyed for preparing tissue culture, molecular biology and protein biochemistry tools for this project; Dr. Hailong Yu for helpful inputs in biochemistry and electrophysiology, and all members of the Wang laboratory for helpful discussions. Cryo-EM data was collected at the University of Texas Southwestern Medical Center Cryo-EM Facility, which is funded by the CPRIT Core Facility Support Award RP170644. This work is supported by NIH grant 1R35GM146860 and the McKnight Scholar Award to W.W.

## Author contributions

X.L. performed the experiments and analyzed the data. W.W. collected the cryo-EM data and reconstructed density maps. X.L. and W.W. designed the project and wrote the manuscript.

## Competing interests

The authors declare no competing interests.
