## [Peer Review File · Nature Communications]

REVIEWER COMMENTS

Reviewer #1 (Remarks to the Author):

In this paper, the authors report four cryo-EM structures of the heteromeric alpha1beta GlyR recombinantly expressed in cell lines and truncated from a large part of the intracellular domain. A previous study of the corresponding author showed a similar work on the heteromeric alpha2beta receptor. However, the new structures present distinctive features, in particular a marked asymmetry in the bottom part of the transmembrane domain, as well as the capture of the receptor in an open and an expanded open state. Authors also claim that expanded open and desensitized states are only partially bound to the agonist glycine. They conclude that these data give insights into the cooperative structural motions mediating gating, and provide a rationale for the gating mechanism the receptor undergoes in more cellular context when bound to the scaffolding protein gephyrin.

The new structures provide interesting information about the conformational landscape of the protein. However, the data interpretation concerning agonist binding site occupancy remains questionable. In addition, the conclusions are often speculative, and the claims not supported by the data. The limitation of the work, especially concerning the use of detergents is not discussed. The work needs clarifications in several important aspects before manuscript assessment.

Major points:

1/ The major concern is the claim that, in the expanded open and desensitized states, only partial glycine binding occurs. In addition, the empty binding sites display the very same local conformation as the glycine-bound sites, meaning they are organized in the high affinity conformation. The claim that high affinity binding sites are empty in the presence of a very high and saturating 2 mM concentration of glycine does not make sense. Could it be that, considering the moderate resolution of the cryo-EM structures (it is interesting to note that the expanded open and desensitized state display the lower resolution of the series, 3.9 and 4.1 angstrom), and considering the small size of the glycine molecule, the density of glycine has been missed at some interfaces in the electron density map, and that those conformations are actually fully occupied by the agonist?

Still, if the conclusion that partial occupancy is correct, then the binding sites at different interfaces will be predicted to contribute differently to desensitization. This should be reflected in the electrophysiological studies. In particular, the claim that, in the desensitized state, no glycine molecules are bound at the interface with the beta subunit should have implications on the desensitization profiles when mutating the sites at beta-alpha and alpha-beta versus alpha-alpha interfaces. The rather limited mutational/electrophysiological data presented in the paper should be analyzed in more details concerning the kinetics and extent of desensitization, and other mutants on the plus and minus sides of the interfaces should be tested to document the observation.

2/ In previous structural studies of the GlyR, especially by the Gouaux lab, the receptor was found highly sensitive to the membrane environment. The receptor adopts different conformations depending on its preparation in detergent, nanodiscs or SMA polymers. The previous study by the corresponding author was performed in nanodiscs, while the present work is in detergent. I was surprised that this key aspect of the receptor biochemistry was completely eluded in the manuscript, and reference to detergent preparation is only found in the methods. This important aspect must be presented and discussed in the main text. Author should justify why detergent has been used in the present work, and the limitation of the work concerning membrane-inserted receptor in physiological conditions.

3/ Concerning the asymmetry of the TMD in the open structure, the authors propose that some charge residues, located in the upper part of the TMD, contribute to rigidify the helices at the alpha-beta interface but promote repulsion at the alpha-alpha interfaces. However, the main asymmetric movements are seen at the lower, and not the upper part of the channel. Author should describe the mechanism in more detail and explain how attraction and repulsion contribute to the local structure and how these constraints are transmitted to the bottom of the receptor. In this line, repulsion between alphas should render the homopentameric receptor prone to expanded opening, is it the case?

4/ a claim of the paper is that the cryo-EM structures show molecular mechanisms that are relevant to the receptor at synapses when aggregated by gephyrin. This is highlighted in the abstract (“These findings point to a gating mechanism that is distinct from homomeric receptors but more compatible with heteromeric GlyRs being clustered at synapses through β subunit–scaffolding protein interactions”) at the end of the results (line 65 “This mechanism ensures that the clustering of GlyRs at synapses through its β subunit does not cause aberrant channel activation independent of glycine”) and in the discussion (line 233 “when β subunit experiences force originating from relative motion with respect to the tethered scaffold...”)

The tighter TMD interaction between the beta and alpha subunits is an interesting information. However, the contribution of this observation to what happens in cell is completely speculative. In addition, gephyrin interacts with the beta subunit at the level of the flexible intracellular loop, and no evidence, to my knowledge, suggest that gephyrin binding alter the receptor function in any way. The idea that gephyrin applies some kind of force to the receptor is pure speculation, and shouldn't be put forward in the abstract and at the end of the introduction.

5/ Annotation of the open state. An important finding of the paper is the structural characterization of a heteromeric GlyR in the open state. However, solely looking at the pore radius does not demonstrate that the channel is conductive to ions, and the state of the art technique is now to perform computational electrophysiology and look for spontaneous chloride translocation events during MD simulations. Authors should at least compare in a supplementary figure their open state with the ones validated as conductive for homopentameric GlyRs. In Fig2f, by eye, it is not clear that the local conformation at -2' is wider than that of the desensitized state.

Minor comments

- Line 87,88, reference to figures seems not correct
- Supp fig 4 g,h: alphaE should be betaE
- line 247 correction of "bule" to "bleu", "gray" to "grey".
- line 252 correction of "gray" to "grey".
- line 270 correction of "gray" to "grey".

Reviewer #2 (Remarks to the Author):

In their new manuscript 'Asymmetric gating of a human hetero-pentameric glycine receptor,' Liu and Wang propose a new structural mechanism for function in pentameric ligand-gated ion channels. This paper appears to build directly on the authors' own previous work (over)expressing and determining cryo-EM structures of modified human lipid nanodisc-embedded $\alpha 2\beta$ GlyRs in presumed nonconducting states, and that of others including the Gouaux group on structures of eukaryotic $\alpha 1$ GlyRs with and without β subunits from heterologous and native sources in lipid nanodiscs in various states.

The four new structures presented here, all of modified recombinant human $\alpha 1\beta$ GlyRs in detergent, include for apparently the first time heteromeric channels in states consistent with ion conduction. Interestingly, the pore profile of one of these is roughly comparable to previous open structures of homomeric $\alpha 1$ GlyRs from zebrafish, while another, wider state is reminiscent of 'expanded open' structures of the same. However, both potentially conductive states are notably asymmetric, particularly near the cytoplasmic side. Based on these findings, the authors propose a provocative gating mechanism for heteromeric channels involving subunit-specific rearrangements in the transmembrane domain.

Aside from some grammatical errors, the manuscript is concise and compelling. The structures support the growing consensus on stoichiometry for heteromeric GlyRs, and offer new testable hypotheses to populate the gating landscape. In support of the structures, functional data convincingly establish the limited functional impact of cryo-EM modifications (at least in HEK cells) and distributed contributions of agonist binding at both symmetric and asymmetric interfaces. On the other hand, some of the mechanistic claims do not seem fully justified, particularly in context of their deviations from previous reports and models in both the extracellular and transmembrane domains, and in absence of functional or computational validation. These interesting structures should surely be made available to the scientific community, with appropriate revision to the narrative.

Major comments:

1. More detailed comparisons, e.g. with statistical deviations, with aligned regions of previously reported apo, open, expanded-open, and desensitized states could be helpful in understanding the variability as well as conservation of the present structures. Is the apo state entirely comparable to previously reported closed channels? A radius of 1.8 Å at the hydrophobic gate seems relatively large compared to both heteromeric and homomeric GlyRs; conversely, the constriction seems unusually distributed along a ~10-Å swath of the pore axis, rather than localized around the 9' sidechain. Can this be attributed to subtype specificity, or to intermediate resolution?

2. What can account for the absence of Gly in multiple sites in the expanded-open and desensitized states – where it has previously been observed in closely related constructs? Could differences in experimental conditions account for partial Gly dissociation in comparison to the previously reported desensitized a1b GlyR? Is it possible that the relatively low resolution of the present structures precludes modeling of agonist molecules that should nonetheless be presumed present, as in previously reported GlyR-Gly structures in detergent? This seems especially important to discuss given that the structures with fewer ligands are reported to lower overall resolution (3.9–4.1 Å) than the fully occupied presumed open state (3.6 Å), and that little or no structural rearrangement is observed in the presumed unoccupied sites -- in contrast to unoccupied sites in previously reported heteromeric nAChRs and GABA(A)Rs. Is the resolution sufficient to determine whether any sidechains, particularly in the aromatic box, reorient in the absence of Gly? If the modeled occupancies do accurately report the state of experimental particles, it is tempting to speculate that asymmetric deformation of the transmembrane domain – possibly as an artefact of detergent solubilization – might allosterically influence ligand binding. Conversely, what could be the mechanistic relevance of sub-maximal Gly occupancy in driving alternative pore conformations?

3. Whereas asymmetry is not inherently surprising for a heteromeric receptor, the proposed conductive structures appear to feature dramatic and variable displacement of isolated transmembrane regions near the (deleted) intracellular domain, creating gaps of up to 16 Å between proximal pore-lining helices. What would be the consequences of this unanticipated deformation for solvent interactions? What is the buried surface area of the various subunits, particularly in this region? Might portions of the helical interfaces be exposed to the conduction pathway and/or lipid bilayer? Molecular simulations, even on short timescales, might be particularly informative as to the solvation of these structures, as well as their metastability.

4. The electrostatic rationale offered for disruption of helical interfaces is intriguing. How can repulsion between a1 subunits at the level of 19' account for such dramatic and seemingly localized displacement at -2' – despite having no such apparent effect on homomeric channels? Whereas this proposal is consistent with the previously well documented influence of R271, no further transmembrane

mutations appear to be characterized or described in this work; if truly important in gating, surely such dramatic rearrangements should be influenced by more local mutations near the intracellular side?

5. The possible influence of detergent conditions should be stated more explicitly. It seems particularly important to recognize that at least some of the present structures are reminiscent of hyperexpanded or collapsed pores previously reported in detergent-bound GlyRs or GABA(A)Rs respectively; for both these channel types, reconstitution in lipid nanodiscs appeared to be critical to stabilize pores more consistent with function and simulations. Indeed, in the widest state, the radius at -2' appears to be even more expanded than other expanded-open structures in lipids, more consistent with the homomeric $\alpha 1$ GlyR-Gly structure in detergent. What can the authors speculate to be the relevance, if any, of this expanded open state?

6. The final mechanistic proposal involving β -subunit interactions with scaffolding proteins (Fig 5), particularly the notion that asymmetric subunit contacts ensure 'GlyR responds as a rigid body' to scaffold-related forces, is interesting but not directly supported by experiments in this work – particularly given that the region implicated in scaffold binding is largely absent. It is surely worth mentioning these ideas in discussion, but it seems a stretch to include them explicitly in the summary, final figure, etc.

Minor comments:

1. Given the seemingly moderate resolutions of the maps, and surprisingly dramatic transitions in the inner pore, it could help to have access to original maps and models.
2. Were proteins expressed in HEK293S GnTI- cells, as in the authors' previous work?
3. In Fig 1, several lipids and glycans are shown, but not clearly described in the text. Please clarify also which 'complex with glycine' is shown in panel c.
4. In Fig 2, representations of the open state seem strangely inconsistent with the other three states, making direct comparisons challenging: why show chain B here instead of chain C?
5. In Fig 3h, the β subunit appears to transition in an opposite fashion (counter-clockwise rather than clockwise) compared to either depicted $\alpha 1$ subunit – contrary to the direction of the drawn arrows, and to apparent shifts in Fig 4b. Please check and/or clarify.
6. In Fig 4, how do the surfaces in panels g–j correspond to the helix depictions in panel f? It would likely be most informative if the entire area in f can be shown in g–j.

Reviewer #1 (Remarks to the Author):

In this paper, the authors report four cryo-EM structures of the heteromeric alpha1beta GlyR recombinantly expressed in cell lines and truncated from a large part of the intracellular domain. A previous study of the corresponding author showed a similar work on the heteromeric alpha2beta receptor. However, the new structures present distinctive features, in particular a marked asymmetry in the bottom part of the transmembrane domain, as well as the capture of the receptor in an open and an expanded open state. Authors also claim that expanded open and desensitized states are only partially bound to the agonist glycine. They conclude that these data give insights into the cooperative structural motions mediating gating, and provide a rationale for the gating mechanism the receptor undergoes in more cellular context when bound to the scaffolding protein gephyrin.

The new structures provide interesting information about the conformational landscape of the protein. However, the data interpretation concerning agonist binding site occupancy remains questionable. In addition, the conclusions are often speculative, and the claims not supported by the data. The limitation of the work, especially concerning the use of detergents is not discussed. The work needs clarifications in several important aspects before manuscript assessment.

Major points:

Reviewer 1 comment 1:

1/ The major concern is the claim that, in the expanded open and desensitized states, only partial glycine binding occurs. In addition, the empty binding sites display the very same local conformation as the glycine-bound sites, meaning they are organized in the high affinity conformation. The claim that high affinity binding sites are empty in the presence of a very high and saturating 2 mM concentration of glycine does not make sense. Could it be that, considering the moderate resolution of the cryo-EM structures (it is interesting to note that the expanded open and desensitized state display the lower resolution of the series, 3.9 and 4.1 angstrom), and considering the small size of the glycine molecule, the density of glycine has been missed at some interfaces in the electron density map, and that those conformations are actually fully occupied by the agonist?

Still, if the conclusion that partial occupancy is correct, then the binding sites at different interfaces will be predicted to contribute differently to desensitization. This should be reflected in the electrophysiological studies. In particular, the claim that, in the desensitized state, no glycine molecules are bound at the interface with the beta subunit should have implications on the desensitization profiles when mutating the sites at beta-alpha and alpha-beta versus alpha-alpha interfaces. The rather limited mutational/electrophysiological data presented in the paper should be analyzed in more details concerning the kinetics and extent of desensitization, and other mutants on the plus and minus sides of the interfaces should be tested to document the observation.

Author response image 1:

Response: We acknowledge that moderate resolutions map reconstructions may lead to missed densities of glycine molecules that are small size. However, we believe it is not likely the case here for the following reasons. *First*, local resolutions around the glycine binding pockets (and most part of ECDs) are similar across all the structures with sidechains well resolved. In the following **Author response image 1**, we show models and maps around these binding sites (same contour level in each map), where densities for glycine differ clearly in the sites denoted as occupied and unoccupied. These have also been added to **revised supplementary Fig. 3**. *Second*, it is not essential to have all 5 sites occupied to activate GlyR. Our electrophysiological data (**revised Fig. 3d** and **Supplementary Fig. 5**) shows that GlyR $\alpha 1\beta$ can be activated with less than 5 allosteric sites with similar max currents. At the same time, an abundance¹⁻⁶ of glycine dose-response curves in available literatures pointing to a Hill slope of around 2~4, further indicating incomplete glycine binding being sufficient for gating. *Third*, ligand being absent in the capped conformation of allosteric sites has also been observed in the closely related GABA_A receptors. Shown in **Author response image 2** are some recently reported^{7,8} structures near the allosteric sites of (a) $\alpha 1\beta 2\gamma 2$ and (b, c) $\alpha 1\beta 3$ GABA_A, in the presence and absence of the agonist GABA. Despite that only the $\beta(+)\alpha(-)$ sites bind GABA, all 5 allosteric sites are in a similar “capped” conformation

(loop C locations indicated by black arrows in panel **a**) corresponding to GABA bound state. Panel **c** shows close-up comparison between GABA-bound (cyan loop C), GABA-free (magenta loop C) and inhibited (grey) states. Since this is very different from the cationic nicotinic acetylcholine receptors where partial agonist binding resulted in asymmetric ECD conformations^{9,10} and Hill slope of ~ 1 during activation^{11,12}, it is likely that cooperative ECD conformational changes are specific to anionic GABA and glycine receptors.

Author response image 2:

Partial occupancy does not necessarily indicate differential contribution of binding sites to activation, especially in this case where the conformational changes in the ECD is cooperative. This is also supported in our electrophysiological data (Fig. 3, Supplementary Fig. 5) showing that mutating different sites have similar effect to glycine EC₅₀. We performed additional electrophysiological experiments and show that (in **revised Fig. 3**, and listed in **Author response table 1** for ease of reference) abolishing the $\alpha(+)\beta(-)$, or $\alpha(-)\beta(+)$ site had similar effect to gating, while abolishing both these sites at β subunit interface still allow the channel to be fully activated, consistent with one desensitized structure having no glycine bound at both β subunit sites.

The desensitized state is unlikely to have causal link with specific binding patterns of the allosteric sites. This is probably most apparent in the fact that ECD conformations are essentially identical in all activated states (open, semi/pre-open, desensitize and expanded-open) for both homomeric^{7,13,14} and heteromeric⁶ GlyRs when activated with the same ligand. Desensitization kinetics (open \rightarrow desensitized) is more likely a property of the TM and is unaffected by allosteric site mutations as long as ECD can be activated as a whole functional unit. In **Author response image 3** we provide additional data supporting this, showing that the extend of desensitization, as well as its kinetics of allosteric site mutants are identical to that of wild type.

The structures we resolved here most likely do not encompass the complete conformational space, but only a subset of possibilities. Other activated conformations with different patterns of glycine occupancy are very likely present, but simply not

captured here due to finite data size. We have added discussion in the main text to highlight this.

Author response image 3:

Author response table 1:

Constructs	Activated binding pocket	EC_{50} (μM)	Fully activated?	Tau time from open to des
$\alpha 1\beta$	All	98 ± 8	Yes	$\tau = 8.8 \pm 3.6\text{s}$
$\alpha 1\beta R86T$	$3\alpha(+)\alpha(-)$ and $\beta(+)\alpha(-)$	154 ± 5	Yes	$\tau = 7.2 \pm 1.2\text{s}$
$\alpha 1\beta Y231A$	$3\alpha(+)\alpha(-)$ and $\alpha(+)\beta(-)$	214 ± 9	Yes	$\tau = 7.2 \pm 1\text{s}$
$\alpha 1\beta R86TY231A$	$3\alpha(+)\alpha(-)$	258 ± 25	Yes	$\tau = 7.2 \pm 0.6\text{s}$
$\alpha 1F207A\beta$	$\beta(+)\alpha(-)$	>800	No	

Reviewer 1 comment 2:

2/ In previous structural studies of the GlyR, especially by the Gouaux lab, the receptor was found highly sensitive to the membrane environment. The receptor adopts different conformations depending on its preparation in detergent, nanodiscs or SMA polymers. The previous study by the corresponding author was performed in nanodiscs, while the present work is in detergent. I was surprised that this key aspect of the receptor biochemistry was completely eluded in the manuscript, and reference to detergent preparation is only found in the methods. This important aspect must be presented and discussed in the main text. Author should justify why detergent has been used in the present work, and the limitation of the work concerning membrane-inserted receptor in physiological conditions.

Response:

This is a technical question, but actually touches on a very critical aspect of membrane protein study that is currently challenging and poorly understood in many cases – interaction with lipids. In short, digitonin is the best condition we identified at present to understand the working mechanism of $\alpha 1\beta$ GlyR. Better lipid mimetics than nanodiscs, such as lipid vesicles, combined with specific types of lipid mixtures/membrane potentials, will be required for better understanding in the future. Following are some details.

Technically, nanodiscs damaged the structural integrity of the $\alpha 1\beta$ GlyR, and digitonin best preserved it in our tested conditions. We have added **supplementary Figure 1f-i**

(and corresponding methods) showing some failed examples compared with digitonin. Following is our rationale of using digitonin.

Detergents and artificial lipid nanodiscs have both been very effective approaches to isolate membrane proteins from cellular membranes for biophysical/biochemical characterizations¹⁵. Detergents have been shown in some cases to distort protein structure, as well as nanodiscs where disc size/interaction of scaffolds with target protein introduces artifacts. Indeed, which provide a more “native” environment is a general question that has not been fully characterized and is usually case-by-case. For homomeric GlyRs, presumably non-native “expanded open” structures have been reported in a detergent (DDM)¹⁴, and a nanodiscs (SMA nanodiscs)¹⁶ condition, both of which, at the same time, yielded structures that are consistent with channel functions. In many cases the crucial aspect is to identify, among a good variety of detergents, scaffolds and reconstitution techniques, the suitable approach(es) for the specific protein under investigation. The detergent digitonin that we used here is widely believed very gentle and preserves structural lipids that stabilize native protein structures. It is a most used (the most used by 2021¹⁵) condition for membrane protein structure determination. Of course, much better lipid environment mimetics such as lipid vesicles, is likely the future for membrane protein structure determination *in vitro*¹⁷. Whether conformation of GlyR will be different in lipid vesicles comprising different lipid mixtures remain to be discovered. And we have added a section in the main text to discuss this.

Reviewer 1 comment 3:

3/ Concerning the asymmetry of the TMD in the open structure, the authors propose that some charge residues, located in the upper part of the TMD, contribute to rigidify the helices at the alpha-beta interface but promote repulsion at the alpha-alpha interfaces. However, the main asymmetric movements are seen at the lower, and not the upper part of the channel. Author should describe the mechanism in more detail and explain how attraction and repulsion contribute to the local structure and how these constrains are transmitted to the bottom of the receptor. In this line, repulsion between alphas should render the Homopentameric receptor prone to expanded opening, is it the case?

Response:

During channel opening, TM2-TM3 in a large part experienced a rotating motion pivoting on the TM2-TM3 loop. The lower part is more distant from the pivot point and thus experienced more displacement, whereas the upper part is closer to the pivot and moved less. We show this in the **revised Figure 5a**.

The relevance of expanded open state to channel function is currently unclear, and thus it is not straight-forward to test whether homomeric channels are more prone to expanded opening. Based on currently available expanded open structures, where homomeric channels did not show much increased inter-subunit distance^{7,14}, we speculate that repulsion is not likely related to expanded open in homomeric GlyRs.

Reviewer 1 comment 4:

4/ a claim of the paper is that the cryo-EM structures show molecular mechanisms that are relevant to the receptor at synapses when aggregated by gephyrin. This is highlighted in the abstract (“These findings point to a gating mechanism that is distinct from homomeric receptors but more compatible with heteromeric GlyRs being clustered at synapses through β subunit–scaffolding protein interactions”) at the end of the results (line 65 “This mechanism ensures that the clustering of GlyRs at synapses through its β subunit does not cause aberrant channel activation independent of glycine”) and in the discussion (line 233 “when β subunit experiences force originating from relative motion with respect to the tethered scaffold... ”)

The tighter TMD interaction between the beta and alpha subunits is an interesting information. However, the contribution of this observation to what happens in cell is completely speculative. In addition, gephyrin interacts with the beta subunit at the level of the flexible intracellular loop, and no evidence, to my knowledge, suggest that gephyrin binding alter the receptor function in any way. The idea that gephyrin applies some kind of force to the receptor is pure speculation, and shouldn't be put forward in the abstract and at the end of the introduction.

Response:

We are excited by the tighter TMD interactions between the β and the α subunits, which is conceptually consistent with β subunit being the tethering site in its physiological settings. We agree that more rigorous testing will be required to establish this correlation and have removed relevant description in the abstract and introduction, raising this possibility only briefly in the discussion section.

Reviewer 1 comment 5:

5/ Annotation of the open state. An important finding of the paper is the structural characterization of a heteromeric GlyR in the open state. However, solely looking at the pore radius does not demonstrate that the channel is conductive to ions, and the state of the art technique is now to perform computational electrophysiology and look for spontaneous chloride translocation events during MD simulations. Authors should at least compare in a supplementary figure their open state with the ones validated as conductive for homopentameric GlyRs. In Fig2f, by eye, it is not clear that the local conformation at -2' is wider than that of the desensitized state.

Response:

We agree that MD simulation could be a powerful technique for understanding the conduction of ion channels. At the same time, effective simulations usually require careful tuning of methods, timing etc., and in many cases the use of non-physiological conditions/assumptions, to recapitulate experimental observations in multiple ion channel families^{18,19}. Effective and bias-free simulation is still an active field of research^{19,20} and hopefully will become easily reliable in the near future with recent fast developments in computer technologies.

Luckily, MD simulation has been reported with homomeric GlyRs regarding conduction states. As suggested, we compared our $\alpha 1\beta$ GlyR open state structure with

a few structures that have been evaluated using simulation: open state (PDB ID: 6UD3), apo state (PDB ID: 6UBS) and desensitized state (PDB ID: 6UBT) of the homomeric $\alpha 1$ GlyR. In addition, we also compared our structure with recent structure annotated as open based on pore properties (without simulation) (PDB ID: 6PM6). The pore geometry and conformations of the 9' and -2' gates are shown in revised **supplementary figure 6k, l**. Apparently, the pore geometry of $\alpha 1\beta$ GlyR open state resembles both the open state structures, with radii within the expected range based on electrophysiology measurements.

Minor comments

We appreciated very much the reviewer catching the following typos/mistakes and have corrected them accordingly.

-Line 87,88, reference to figures seems not correct

We have corrected the reference.

-Supp fig 4 g,h: αE should be βE

This has been corrected.

- line 247 correction of "bule" to "bleu", "gray" to "grey".

These has been corrected.

- line 252 correction of "gray" to "grey".

This has been corrected.

- line 270 correction of "gray" to "grey".

This has been corrected.

Reviewer #2 (Remarks to the Author):

In their new manuscript 'Asymmetric gating of a human hetero-pentameric glycine receptor,' Liu and Wang propose a new structural mechanism for function in pentameric ligand-gated ion channels. This paper appears to build directly on the authors' own previous work (over)expressing and determining cryo-EM structures of modified human lipid nanodisc-embedded $\alpha 2\beta$ GlyRs in presumed nonconducting states, and that of others including the Gouaux group on structures of eukaryotic $\alpha 1$ GlyRs with and without β subunits from heterologous and native sources in lipid nanodiscs in various states.

The four new structures presented here, all of modified recombinant human $\alpha 1\beta$ GlyRs in detergent, include for apparently the first time heteromeric channels in states consistent with ion conduction. Interestingly, the pore profile of one of these is roughly comparable to previous open structures of homomeric $\alpha 1$ GlyRs from zebrafish, while another, wider state is reminiscent of 'expanded open' structures of the same. However, both potentially conductive states are notably asymmetric, particularly near the cytoplasmic side. Based on these findings, the authors propose a provocative gating

mechanism for heteromeric channels involving subunit-specific rearrangements in the transmembrane domain.

Aside from some grammatical errors, the manuscript is concise and compelling. The structures support the growing consensus on stoichiometry for heteromeric GlyRs, and offer new testable hypotheses to populate the gating landscape. In support of the structures, functional data convincingly establish the limited functional impact of cryo-EM modifications (at least in HEK cells) and distributed contributions of agonist binding at both symmetric and asymmetric interfaces. On the other hand, some of the mechanistic claims do not seem fully justified, particularly in context of their deviations from previous reports and models in both the extracellular and transmembrane domains, and in absence of functional or computational validation. These interesting structures should surely be made available to the scientific community, with appropriate revision to the narrative.

Major comments:

Reviewer 2 comment 1:

1. More detailed comparisons, e.g., with statistical deviations, with aligned regions of previously reported apo, open, expanded-open, and desensitized states could be helpful in understanding the variability as well as conservation of the present structures. Is the apo state entirely comparable to previously reported closed channels? A radius of 1.8 Å at the hydrophobic gate seems relatively large compared to both heteromeric and homomeric GlyRs; conversely, the constriction seems unusually distributed along a ~10-Å swath of the pore axis, rather than localized around the 9' sidechain. Can this be attributed to subtype specificity, or to intermediate resolution?

Response:

This is a very good suggestion to statistically evaluate the variability/conservation of structures in each functional state. This is because structures assigned to specific functional states fundamentally represent snapshots of enriched conformations under certain biochemical conditions. Clearly there are differences among structures annotated in the same functional states. Since the 9' and -2' gates constitute the constrictions that determine conduction capabilities, we analyzed the pore radii at these gates of available glycine receptor structures and included them in **revised supplementary table 2**. Constriction of ~1.8 Å is indeed larger than known apo/resting state structures, where 1.4 ~ 1.5 (1.43 ± 0.05) Å are typical. We further refined structural details in the pore areas against known apo/resting structures and have updated our atomic model, where the constriction is ~1.5 Å. This is similar to that of α2β GlyR and slightly larger than 1.4 Å in homomeric GlyRs, indicating that the 0.2 ~ 0.3 Å variance arises from both limited resolutions, and partly from differences in subtypes.

Regarding the open, expanded open and desensitized states, the geometry of the pore, the minimal radii and the position where they reside in the pore is typical of what have been reported in available structures. Please see **supplementary table 2** for details.

Reviewer 2 comment 2:

2. *What can account for the absence of Gly in multiple sites in the expanded-open and desensitized states – where it has previously been observed in closely related constructs? Could differences in experimental conditions account for partial Gly dissociation in comparison to the previously reported desensitized $\alpha 1\beta$ GlyR? Is it possible that the relatively low resolution of the present structures precludes modeling of agonist molecules that should nonetheless be presumed present, as in previously reported GlyR-Gly structures in detergent? This seems especially important to discuss given that the structures with fewer ligands are reported to lower overall resolution (3.9–4.1 Å) than the fully occupied presumed open state (3.6 Å), and that little or no structural rearrangement is observed in the presumed unoccupied sites -- in contrast to unoccupied sites in previously reported heteromeric nAChRs and GABA(A)Rs. Is the resolution sufficient to determine whether any sidechains, particularly in the aromatic box, reorient in the absence of Gly? If the modeled occupancies do accurately report the state of experimental particles, it is tempting to speculate that asymmetric deformation of the transmembrane domain – possibly as an artefact of detergent solubilization – might allosterically influence ligand binding. Conversely, what could be the mechanistic relevance of sub-maximal Gly occupancy in driving alternative pore conformations?*

Response:

This is an important question that has also been raised by **reviewer 1 in comment 1**. In this case, we do not think partial occupancy is a result of lower resolutions in this case. Briefly, map quality clearly shows differences in glycine density in occupied and non-occupied sites. There are also subtle differences in side-chain conformations, but we are not confident in interpretation due to the limited resolution. We also performed additional electrophysiological experiments showing that partial occupancy is sufficient to activate GlyR. Due to the cooperativity of the extracellular domain, it is unlikely that sub-maximal glycine occupancy is causally linked to specific pore conformations. Also, we note that the partial occupancy structures that we resolved here is very likely only a subset in the possible conformational space, where other patterns of partial occupancy may lead to the same functional states. Compared to GABA_A and nAChR receptors, the unoccupied sites in GABA_A receptors actually resemble that of occupied, where clear asymmetry is observed in nAChRs. This coincides with an activation Hill slope of ~2-4 for GlyR and ~1 for nAChR activation^{11,12}, indicating cooperative ECD conformational change is not conserved across all Cys-loop receptors. Please also **see response to reviewer 1 comment 1** for more details.

We do believe that differences in experimental conditions is related to why we were able to observe the partially occupied conformations. The reported $\alpha 1\beta$ (more precisely $\alpha 1\beta$ and $\alpha 2\beta$ mixture, with $\alpha 1\beta$ being dominant) GlyR structure (PDB:5BKG) was resolved in the presence of an antibody that directly binds to the allosteric sites, locking them in the glycine-bound conformation²¹. In addition, higher concentration of glycine (10 mM) was used. In this study, no antibody was used, together with a lower glycine concentration (2 mM).

We do not think asymmetric transmembrane domain conformations are artefacts arising from detergent solubilization for the following reasons. **First**, both “good” and “distorted” GlyR structures have been resolved in both detergents and nanodiscs^{13,14,16,22}. Please also see **response to Reviewer 1 comment 2** for more detailed discussion on detergents/membrane mimetics. In addition, asymmetric TM conformation has also been observed in $\alpha 2\beta$ GlyR structures that was determined using nanodiscs (7KUY, 5GKB). **Second**, we performed additional experiments to show that widening of subunit distance, which results in asymmetric conformation, indeed underly $\alpha 1\beta$ GlyR opening. Please also see **response to comment 4** for details.

In short, glycine receptor activation does not require the glycine binding pocket to be fully occupied, which is consistent with partially occupied binding pockets in activated structures of $\alpha 1\beta$ GlyR.

Reviewer 2 comment 3:

3. Whereas asymmetry is not inherently surprising for a heteromeric receptor, the proposed conductive structures appear to feature dramatic and variable displacement of isolated transmembrane regions near the (deleted) intracellular domain, creating gaps of up to 16 Å between proximal pore-lining helices. What would be the consequences of this unanticipated deformation for solvent interactions? What is the buried surface area of the various subunits, particularly in this region? Might portions of the helical interfaces be exposed to the conduction pathway and/or lipid bilayer? Molecular simulations, even on short timescales, might be particularly informative as to the solvation of these structures, as well as their metastability.

Response:

This is a very good question, and we believe is related to **reviewer 2 comment 5** regarding the physiological/biochemical relevance of “expanded-open” conformation. Shown in **Author response image 3** are buried areas between subunits TM of all 4 structures. Clearly in both the expanded open and open states, buried area significantly decreased between the two TM that moved away from each other.

Author response image 3:

Related to this, we see lipid-like density filling in the gaps especially in the expanded open state, where extended density partially occludes the pore. In the open state, lipid densities are found between the widened subunit interfaces, away from the pore (**Author response image 4**). Based on these experimental observations, we believe both the “expanded open” and open conformations are partially stabilized by structural

lipids at inter-subunit interfaces where subunits swayed apart. It is also tempting to speculate that the “expanded open” state is not conductive in lipid membranes. Please also see **response to reviewer 2 comment 5** for more discussion of this conformation. We agree that MD simulation could provide valuable information regarding the conduction and stability of these conformations. However, considering the parameter space to explore in MD simulation for reliable interpretation, we think it would be a great future project, combined with functional/microscopy experiments and structures determine in lipid vesicles, more fully understand the structural landscape of GlyR gating.

Author response image 4:

Reviewer 2 comment 4:

4. The electrostatic rationale offered for disruption of helical interfaces is intriguing. How can repulsion between $\alpha 1$ subunits at the level of 19' account for such dramatic and seemingly localized displacement at -2' – despite having no such apparent effect on homomeric channels? Whereas this proposal is consistent with the previously well documented influence of R271, no further transmembrane mutations appear to be characterized or described in this work; if truly important in gating, surely such dramatic rearrangements should be influenced by more local mutations near the intracellular side?

Response:

How charge at 19' lead to more extensive widening at -2' is apparent in the structures, which we did not present clearly. This has been raised by both reviewers (please also see **reviewer 1 comment 3**) and we appreciate very much the reviews' comment to better present this. We have added an additional main figure, **revised Figure 5**, to highlight this. Briefly, the displacement at 19' is not as extensive at -2' is resulting from a rotational motion pivoting around the M2-M3 loop. 19' is closer to the pivot point and thus moves less compared to -2' that is further from pivot. We speculate that homomeric GlyRs open in a symmetric manner instead of a local widening due to their

symmetry – there is no “weak” point in the TM for focused conformational change. Whereas in heteromeric channels, differences in subunit interfaces renders weak points for local deformations. We presented this movement in **revised Figure 5a**.

Mutations near -2' is a very good suggestion to test the asymmetric open mechanism. We have performed the experiments and included the outcomes in **revised figure 5**. Briefly, crosslinking between adjacent subunits near -2' site through disulfide bonds (preventing widening of inter-subunit distance) significantly decreased GlyR activity, which is reversible upon application of reducing reagent DTT. These experiments demonstrate the importance of inter-subunit distance increase in heteromeric channel opening.

Reviewer 2 comment 5:

5. The possible influence of detergent conditions should be stated more explicitly. It seems particularly important to recognize that at least some of the present structures are reminiscent of hyperexpanded or collapsed pores previously reported in detergent-bound GlyRs or GABA(A)Rs respectively; for both these channel types, reconstitution in lipid nanodiscs appeared to be critical to stabilize pores more consistent with function and simulations. Indeed, in the widest state, the radius at -2' appears to be even more expanded than other expanded-open structures in lipids, more consistent with the homomeric $\alpha 1$ GlyR-Gly structure in detergent. What can the authors speculate to be the relevance, if any, of this expanded open state?

Response:

We have added in the abstract and main text to describe the digitonin detergent that we used. As we discussed in **response to reviewer 2 comment 2** and to **reviewer 1 comment 2**, expanded open states has been observed in both detergents¹⁴ and SMA nanodiscs¹⁶. We think the preservation of structural lipids, instead of the specific biochemical methods, is probably more relevant to retaining native channel properties¹⁵. For instances, using mild detergents like digitonin could prove quite effective in preserving native lipids and retaining membrane protein function and structure. Limitation in nanodisc size, interaction with scaffold, or ineffective lipid incorporation might also lead to artefacts. In the case of $\alpha 1\beta$ GlyR, multiple trails for reconstitution of nanodiscs was not successful, indicating either the reconstitution process was too harsh for this protein, or the nanodiscs failed to maintain structural integrity (same methods worked well for $\alpha 2\beta$ GlyR). In the future, better membrane mimetics, such as lipid vesicles with specific lipid mixtures, will be fundamental for understanding protein-lipid interactions¹⁷.

The relevance of expanded open state is a very interesting question and we appreciate the reviewer for raising it. The minimal pore radius of $\alpha 1\beta$ expanded-open structure is closer to that of the homomeric $\alpha 1$ GlyR in SMA nanodiscs (PDB ID: 6PM4) than in detergent (3AJE), with -2' radius in between these two. A plot of pore radii showing comparison has been added in **revised supplementary figure 6l, m**. Interestingly, such “expanded open” states are only reported in conditions that easily allow variable TM sizes – detergents and SMA nanodiscs. Current consensus is that these conformations

are artefacts because the pore radii are larger than the upper limited predicted through conductance measurements of anions with different sizes. However, in expanded open state, we see clear densities of lipids-like materials filling the gap between the 2 subunits, and partially obstructing the conduction pore. It is tempting to speculate that the “expanded open” states might not be conductive, but instead represent a non-conductive state in the glycine-bound state and correspond to the slow recovery (minutes, data not shown) from desensitized state. However, clearly much more testing is required to establish the correlation. In particular, determination of GlyR structures in lipid vesicles might provide important information regarding the functional relevance of “expanded-open” structures.

Reviewer 2 comment 6:

6. The final mechanistic proposal involving b-subunit interactions with scaffolding proteins (Fig 5), particularly the notion that asymmetric subunit contacts ensure ‘GlyR responds as a rigid body’ to scaffold-related forces, is interesting but not directly supported by experiments in this work – particularly given that the region implicated in scaffold binding is largely absent. It is surely worth mentioning these ideas in discussion, but it seems a stretch to include them explicitly in the summary, final figure, etc.

Response: It was an exciting assumption for us, and we agree much more testing is required to establish the correlation. The related sentences in summary and introduce have been deleted. And scaffold protein in final figure has been removed.

We appreciate the following minor comments very much, which made the manuscript clearer and more accurate.

Minor comments:

1. Given the seemingly moderate resolutions of the maps, and surprisingly dramatic transitions in the inner pore, it could help to have access to original maps and models.
We have provided models and corresponding density maps with the revised submission.

2. Were proteins expressed in HEK293S GnTI- cells, as in the authors’ previous work?
Yes, proteins were expressed in HEK293S GnTI- cells.
We highlighted this in the method section.

3. In Fig 1, several lipids and glycans are shown, but not clearly described in the text. Please clarify also which ‘complex with glycine’ is shown in panel c.
The description of lipids and glycan has been added in figure legend of panel c.

4. In Fig 2, representations of the open state seem strangely inconsistent with the other three states, making direct comparisons challenging: why show chain B here instead of chain C?

The orientation is chosen so that movement of helices are most easily appreciated. The B chain moved away from the pore in the open state.

5. In Fig 3h, the b subunit appears to transition in an opposite fashion (counter-clockwise rather than clockwise) compared to either depicted a1 subunit – contrary to the direction of the drawn arrows, and to apparent shifts in Fig 4b. Please check and/or clarify.

This was a mistake, and we appreciate the review for catching it. We have updated figure 3h (β subunit).

6. In Fig 4, how do the surfaces in panels g–j correspond to the helix depictions in panel f? It would likely be most informative if the entire area in f can be shown in g–j. Fig. 4g-h indeed correlate to the regions that are shown in panel f. To better show this, we have provided zoomed out surface presentations, with dashed boxes added to indicate the area in Fig. 4f.

References

- 1 Schmieden, V., Grenningloh, G., Schofield, P. R. & Betz, H. Functional expression in *Xenopus* oocytes of the strychnine binding 48 kd subunit of the glycine receptor. *The EMBO journal* **8**, 695-700 (1989).
- 2 Bormann, J., Rundstrom, N., Betz, H. & Langosch, D. Residues within transmembrane segment M2 determine chloride conductance of glycine receptor homo- and hetero-oligomers. *The EMBO journal* **12**, 3729-3737 (1993).
- 3 Grudzinska, J. *et al.* The beta subunit determines the ligand binding properties of synaptic glycine receptors. *Neuron* **45**, 727-739, doi:10.1016/j.neuron.2005.01.028 (2005).
- 4 Durisic, N. *et al.* Stoichiometry of the human glycine receptor revealed by direct subunit counting. *The Journal of neuroscience : the official journal of the Society for Neuroscience* **32**, 12915-12920, doi:10.1523/JNEUROSCI.2050-12.2012 (2012).
- 5 Raltshev, C., Hetsch, F., Winkelmann, A., Meier, J. C. & Semtner, M. Electrophysiological Signature of Homomeric and Heteromeric Glycine Receptor Channels. *The Journal of biological chemistry* **291**, 18030-18040, doi:10.1074/jbc.M116.735084 (2016).
- 6 Yu, H., Bai, X. C. & Wang, W. Characterization of the subunit composition and structure of adult human glycine receptors. *Neuron* **109**, 2707-2716 e2706, doi:10.1016/j.neuron.2021.08.019 (2021).
- 7 Kasaragod, V. B. *et al.* Mechanisms of inhibition and activation of extrasynaptic alphabeta GABA(A) receptors. *Nature* **602**, 529-533, doi:10.1038/s41586-022-04402-z (2022).
- 8 Kim, J. J. *et al.* Shared structural mechanisms of general anaesthetics and benzodiazepines. *Nature* **585**, 303-308, doi:10.1038/s41586-020-2654-5 (2020).
- 9 Rahman, M. M. *et al.* Structural mechanism of muscle nicotinic receptor desensitization and block by curare. *Nature structural & molecular biology* **29**, 386-394, doi:10.1038/s41594-022-00737-3 (2022).
- 10 Gharpure, A. *et al.* Agonist Selectivity and Ion Permeation in the alpha3beta4 Ganglionic Nicotinic Receptor. *Neuron* **104**, 501-511 e506, doi:10.1016/j.neuron.2019.07.030 (2019).
- 11 Watty, A., Methfessel, C. & Hucho, F. Fixation of allosteric states of the nicotinic acetylcholine receptor by chemical cross-linking. *Proceedings of the National Academy of Sciences of the United States of America* **94**, 8202-8207, doi:10.1073/pnas.94.15.8202 (1997).
- 12 Briggs, C. A. & McKenna, D. G. Activation and inhibition of the human alpha7 nicotinic acetylcholine receptor by agonists. *Neuropharmacology* **37**, 1095-1102, doi:10.1016/s0028-3908(98)00110-5 (1998).
- 13 Kumar, A. *et al.* Mechanisms of activation and desensitization of full-length glycine receptor in lipid nanodiscs. *Nature communications* **11**, 3752, doi:10.1038/s41467-020-17364-5 (2020).

- 14 Du, J., Lu, W., Wu, S., Cheng, Y. & Gouaux, E. Glycine receptor mechanism elucidated by electron cryo-microscopy. *Nature* **526**, 224-229, doi:10.1038/nature14853 (2015).
- 15 Choy, B. C., Cater, R. J., Mancina, F. & Pryor, E. E., Jr. A 10-year meta-analysis of membrane protein structural biology: Detergents, membrane mimetics, and structure determination techniques. *Biochim Biophys Acta Biomembr* **1863**, 183533, doi:10.1016/j.bbamem.2020.183533 (2021).
- 16 Yu, J. *et al.* Mechanism of gating and partial agonist action in the glycine receptor. *Cell*, doi:10.1016/j.cell.2021.01.026 (2021).
- 17 Tao, X., Zhao, C. & MacKinnon, R. Membrane protein isolation and structure determination in cell-derived membrane vesicles. *Proceedings of the National Academy of Sciences of the United States of America* **120**, e2302325120, doi:10.1073/pnas.2302325120 (2023).
- 18 Kutzner, C. *et al.* Insights into the function of ion channels by computational electrophysiology simulations. *Biochimica et biophysica acta* **1858**, 1741-1752, doi:10.1016/j.bbamem.2016.02.006 (2016).
- 19 Flood, E., Boiteux, C., Lev, B., Vorobyov, I. & Allen, T. W. Atomistic Simulations of Membrane Ion Channel Conduction, Gating, and Modulation. *Chem Rev* **119**, 7737-7832, doi:10.1021/acs.chemrev.8b00630 (2019).
- 20 Pohorille, A. & Wilson, M. A. Computational Electrophysiology from a Single Molecular Dynamics Simulation and the Electrodiffusion Model. *J Phys Chem B* **125**, 3132-3144, doi:10.1021/acs.jpccb.0c10737 (2021).
- 21 Zhu, H. & Gouaux, E. Architecture and assembly mechanism of native glycine receptors. *Nature*, doi:10.1038/s41586-021-04022-z (2021).
- 22 Huang, X., Chen, H., Michelsen, K., Schneider, S. & Shaffer, P. L. Crystal structure of human glycine receptor- α 3 bound to antagonist strychnine. *Nature* **526**, 277-280, doi:10.1038/nature14972 (2015).

REVIEWER COMMENTS

Reviewer #1 (Remarks to the Author):

The authors did a good job in answering most of my concerns. However, the claim that in expanded and desensitized structures some glycine sites are empty is still surprising to me given the 2 mM concentration of glycine. It is hard to me to be fully convinced from the snapshots of electron densities provided (images are at low resolution). In addition, the authors state in the rebuttal that "There are also subtle differences in side-chain conformations, but we are not confident in interpretation due to the limited resolution". Given that a glycine molecule is more or less the same size than a residue side chain, maybe a direct validation of the electron density map by a cryo-EM specialist (that I am not) would be needed to validate this important point.

Reviewer #2 (Remarks to the Author):

In their revised manuscript "Asymmetric gating of a human hetero-pentameric glycine receptor," Liu and Wang present an evolving narrative regarding their cryo-EM structures of $\alpha 1\beta$ glycine receptors in digitonin. Initial concerns regarding the potential for lipid reconstitution (R1-C2, R2-C5), functional evidence for transmembrane transitions (R1-C3, R2-C3), speculation on interactions with gephyrin (R1-C4, R2-C6), and radius of the closed gate (R2-C1) have been helpfully clarified.

On the other hand, the models and maps now provided make even more clear some fundamental concerns about the provocative proposed gating mechanism. To enable rigorous consideration in the wider community, it is recommended that the authors moderate their annotation, particularly of the structure identified as "the open state." These and other remaining concerns, and associated recommendations, are outlined below.

1) Pore densities: As initially raised in R1-C5 and R2-C3, the most dramatic claim in this manuscript is that the open structure features a large splaying at one (but not the other two) of the α/α transmembrane interfaces, not apparent in the apo or desensitized structures. As now shown in *author response image 3*, the proposed opening transition disrupts 75% of the buried surface area at the splayed (B/C) transmembrane interface. Moreover, as shown in *author response image 4* and the map, the pore appears to be occupied by a branched nonprotein density, extending continuously from the pore-facing T(13') residues to the intracellular side. One branch of this density appears to bridge the B/C interface, separating sidechains of e.g. V(1') from the complementary A(-1'); presumably this is the "structural lipid" referenced in response to R2-C3 (though not in the manuscript text). A second branch of this density appears to block the channel pore, spanning the 6', 2', and -2'

sidechains of all five subunits. It is difficult to conceive of this apparent lipid as purely structural, or to imagine this structure “allows the conduction of Cl⁻” (p. 9).

As noted in response to R2-C5, a similar density occupies the pore and C/D interface in the structure annotated “expanded open.” In this case, the authors explicitly acknowledge “lipids-like materials filling the gap between the 2 subunits, and partially obstructing the conduction pore,” and “speculate that the expanded open states might not be conductive, but instead represent a non-conductive state in the glycine-bound state and correspond to the slow recovery...from desensitized.” It seems strange to acknowledge an apparent lipidic block of the channel pore in this structure (at least in the rebuttal) but not in the open structure. Moreover, these densities are apparent at equal or lower thresholds compared to other nonprotein densities which the authors have built as partial lipids in the same structures; it seems imprecise or even misleading not to mention or show them in the manuscript.

Given these concerns, it is recommended that *author response image 4* (or a similar illustration of pore/interfacial lipidic densities) be included in the manuscript, and described in Results/Discussion.

2) Annotation: Of course, the presence of a lipidic moiety in the channel pore does not preclude its relevance. Previous ligand-gated ion-channel structures have also contained tubular pore densities, particularly in the presence of detergent. It is plausible that lipophilic agents in these structures occupy or even stabilize pre-existing cavities in functionally relevant states. Still, it seems inaccurate to ascribe to this open structure “similar pore geometry” (p. 5) as in previously reported homomeric receptors, or to claim that the “expanded open state pore geometry...shows that they are largely similar” (p. 6). The pore may be similar along its minimal dimension, but it is so expanded along its maximal dimension as to be contiguous with solvent. Structural lipids would presumably be required to stabilize this dissociated subunit interface, while avoiding the ion pathway (at least in the open state). Molecular simulations demonstrating stability and/or conductance of such structures could help to support their relevance, although—per the authors’ response to R1-C5 and R2-C3—configuration of such lipids might be complex. Similarly dissociated transmembrane subunits in a bacterial-homolog structure were previously shown to be unstable in simulations, even with lipids inserted (PMID 36480474). In absence of computational evidence for stability and/or conduction, the structural annotation does not seem clearly justified.

It is recommended that the structures determined with glycine—currently annotated “open/expanded open/desensitized”—should be renamed e.g. “Gly-1/2/3,” with proposals as to their functional state(s) primarily limited to Discussion. If a speculative version of Fig. 6 is kept in the manuscript, it seems appropriate to include a structural lipid in the open state.

3) Mutagenesis: As described in response to R1-C3 and R2-C4 and Fig. 5, it is helpful that the authors show modifications at the α/α interface decrease channel activity. Still, these results do not clearly

support the novel open structure, nor differentiate it from homomeric models. Repulsion at R(19')—the only residue on the outward-facing half of $\alpha 1:M2$ carrying a formal charge—is meant to initiate a pivot motion that dissociates B/C subunits from one another. However, the charge distributed on the complementary face of $\alpha 1$ is relatively weak (Fig. 4j), unlikely to overcome the energetic cost of disrupting van der Waals interactions across $\frac{3}{4}$ of the subunit interface. Repulsion among R(19') residues was previously proposed to promote expansion of the outward-facing end of the homomeric glycine-receptor pore (Ref. 42)—but not to disrupt hydrophobic interfaces distal to the proposed repulsion, nor stimulate insertion of structural lipids. Subsequent structures of homomeric $\alpha 1$ GlyRs (Refs. 11, 12) indicated that R(19') forms state-dependent polar/charged interactions with M1 and M3, and/or with lipophilic modulators such as ivermectin, suggesting multiple mechanisms by which this residue might influence function. Given that neutralization at R(19') is an established loss-of-function mutation in homomeric glycine receptors, possibly to an even greater extent than in heteromers (Ref. 8), this effect cannot substantiate a distinctive mechanism of heteromer gating.

The authors also show that crosslinking $\alpha 1$ subunits at the level of -1'/1'—whose C β distance increases from 4 to 8 Å at one interface in the proposed opening transition—suppresses function. However, given the propensity of crosslinking to select nonphysiological nonfunctional states, these results would be more convincing with appropriate controls. If gating indeed relies on displacement at α/α but not α/β or β/α interfaces (as indicated by the charge-repulsion model), combinations of the $\alpha:A(-1')C$ or $V(1')C$ substitutions with $\beta:V(1')C$ or $A(-1')C$ respectively should be functionally insensitive to oxidation. Alternatively, if asymmetric gating is indeed a distinctive feature of heteromeric receptors, the $\alpha:A(-1')C+V(1')C$ substitution in homomeric receptors should be oxidation-insensitive.

Given the above concerns, it seems important to comment that perturbations at 19' and -1'/1' are consistent with the present structures, but do not clearly prefer them over previously proposed mechanisms. If possible, control experiments mutating β subunits and/or homomeric receptors should be shown.

4) Ligands: In response to R1-C1, the authors argue that “local resolutions around the glycine binding pockets...are similar across all the structures with sidechains well resolved...densities for glycine differ clearly in the sites denoted as occupied and unoccupied.” However, the densities shown in *author response image 1*, *Supplementary Fig. 3c-d*, and the supplied maps do not look so clear. At most of these interfaces, density built as ligand appears not as a discrete volume, but as an extension of neighboring sidechains, potentially interchangeable with alternative rotamers. It does not seem like a coincidence that ligand densities are even less apparent in the lower-resolution structures, and that they do not necessarily overlap built ligands. The point is well taken that occupancy of all possible glycine binding sites is not required for activation, and it is helpful that the authors recognize “other activated conformations with different patterns of glycine occupancy are very likely present.” On the other hand, they acknowledge “there are also subtle differences in side-chain conformations, but we are not confident in interpretation due to the limited resolution” (response to R2-C2); the ligand does not appear any better defined than the surrounding protein. Rather than building ligands at a likely

underdetermined subset of possible sites, it seems more reasonable to presume that all five sites are partially occupied in these classes, with sufficient uncertainty to preclude definitive building.

Even in the fully occupied open state, glycine densities and poses differ between interfaces, and from previous studies. As supported by the authors' own functional data, previous studies have identified electrostatic interactions between R65 and the glycine carboxylate, and a potential cation- π contact between the electronegative face of F207 and the glycine amine. However, in this structure the carboxylate lies $>4 \text{ \AA}$ from the R65 guanidinium in three of five subunits; the glycine amine orients either towards the electropositive edge F207, or away from it entirely. Other glycine contacts appear to be implausibly close, including $<2.5\text{-}\text{\AA}$ intermolecular heavy-atom distances at all interfaces; at the C/D interface, the amine lies just 1 \AA from F159. To be sure, a lack of definitive ligand densities does not negate the value of these structures. However, a more conservative approach to assigning stoichiometry and poses seems in order, especially at resolutions approaching 4 \AA .

Given that the present data do not clearly justify modeling of glycine in some sites versus others, nor physically reasonable modeling of the ligand, it is recommended to remove glycine from the deposited models, and to reserve description of its likely (partially occupied) location(s) for Discussion.

REVIEWER COMMENTS

Reviewer #1 (Remarks to the Author):

The authors did a good job in answering most of my concerns. However, the claim that in expanded and desensitized structures some glycine sites are empty is still surprising to me given the 2 mM concentration of glycine. It is hard to me to be fully convinced from the snapshots of electron densities provided (images are at low resolution). In addition, the authors state in the rebuttal that "There are also subtle differences in side-chain conformations, but we are not confident in interpretation due to the limited resolution". Given that a glycine molecule is more or less the same size than a residue side chain, maybe a direct validation of the electron density map by a cryo-EM specialist (that I am not) would be needed to validate this important point.

Response:

This is indeed puzzling from the perspective of simple binding reactions as 2 mM is around 20 folds of the 100 μ M glycine EC_{50} . At the same time, EC_{50} derived from whole-cell electrophysiological measurements does not necessarily equal binding affinity at each site, especially in this case where 5 sites are capable of binding but less than 5 are required for activation. In addition, it is not uncommon in structure determination where very tight interactions do not result in complete occupancy in resolved density maps. One class of examples is antibody-protein complexes where nano- to pico- molar affinity antibodies show partial occupancy in resolved structures (at micro-molar concentrations) of related ion channels and other receptors.

Determining side-chain location with high accuracy (within fractions of an angstrom) is not quite equivalent to telling whether the sidechain, and similarly a glycine molecule, is present or not. In the structures presented here, density maps are consistent with our assignments of bound glycine. However, to be more conservative, we have removed all glycine atomic models from the map with the lowest overall resolution (desensitized state, 4.1 Å) and included more discussion in the main text about partial occupancy.

Reviewer #2 (Remarks to the Author):

In their revised manuscript "Asymmetric gating of a human hetero-pentameric glycine receptor," Liu and Wang present an evolving narrative regarding their cryo-EM structures of $\alpha 1\beta$ glycine receptors in digitonin. Initial concerns regarding the

potential for lipid reconstitution (R1-C2, R2-C5), functional evidence for transmembrane transitions (R1-C3, R2-C3), speculation on interactions with gephyrin (R1-C4, R2-C6), and radius of the closed gate (R2-C1) have been helpfully clarified.

On the other hand, the models and maps now provided make even more clear some fundamental concerns about the provocative proposed gating mechanism. To enable rigorous consideration in the wider community, it is recommended that the authors moderate their annotation, particularly of the structure identified as “the open state.” These and other remaining concerns, and associated recommendations, are outlined below.

1) Pore densities: As initially raised in R1-C5 and R2-C3, the most dramatic claim in this manuscript is that the open structure features a large splaying at one (but not the other two) of the α/α transmembrane interfaces, not apparent in the apo or desensitized structures. As now shown in *author response image 3*, the proposed opening transition disrupts 75% of the buried surface area at the splayed (B/C) transmembrane interface. Moreover, as shown in *author response image 4* and the map, the pore appears to be occupied by a branched nonprotein density, extending continuously from the pore-facing T(13') residues to the intracellular side. One branch of this density appears to bridge the B/C interface, separating sidechains of e.g. V(1') from the complementary A(-1'); presumably this is the “structural lipid” referenced in response to R2-C3 (though not in the manuscript text). A second branch of this density appears to block the channel pore, spanning the 6', 2', and -2' sidechains of all five subunits. It is difficult to conceive of this apparent lipid as purely structural, or to imagine this structure “allows the conduction of Cl⁻” (p. 9).

As noted in response to R2-C5, a similar density occupies the pore and C/D interface in the structure annotated “expanded open.” In this case, the authors explicitly acknowledge “lipids-like materials filling the gap between the 2 subunits, and partially obstructing the conduction pore,” and “speculate that the expanded open states might not be conductive, but instead represent a non-conductive state in the glycine-bound state and correspond to the slow recovery...from desensitized.” It seems strange to acknowledge an apparent lipidic block of the channel pore in this structure (at least in the rebuttal) but not in the open structure. Moreover, these densities are apparent at equal or lower thresholds compared to other nonprotein densities which the authors have built as partial lipids in the same structures; it seems imprecise or even misleading not to mention or show them in the manuscript.

Given these concerns, it is recommended that *author response image 4* (or a similar

illustration of pore/interfacial lipidic densities) be included in the manuscript, and described in Results/Discussion.

Response:

We much appreciate that the reviewer bringing this topic up. Indeed, these non-protein densities are very interesting to us. The reason that we did not discuss them in more detail in the previous version is because we could only provide our best speculations of their identity and function in the absence experimental support. Densities near or in the pore are quite commonly found in ion channel structures and usually not modeled likely for similar reasons. Since this is indeed interesting to the reader, we agree with the reviewer that we should include the discussion in the main text. For this, we have added supplementary figure 7, a remake of author response image 4 with colored densities and less crowding, highlighting these densities, and discussed how these densities may related to GlyR function in more details in the discussion section. Briefly, much more extensive densities both at subunit interfaces and in the pore are observed in the expanded-open state compared to the open state. This indicates that these substances, presumably lipids, are required to physically stabilized the expanded-open structure and probably create a slow recovering non-conductive state since the recovery requires dissociation of these lipids. In contrast, in the open state, non-protein densities at the subunit interface are much weaker and scattered, indicating weak/non-stable occupancy. The densities in the pore are also weaker and more resemble a beads-on-a-string shape that breaks up into beads at higher thresholds, reminiscent of less-resolved ionic densities. It remains possible that these densities represent lipid tails entering the pore, stabilizing the meta-stable open state, or in the unlikely case, contribute to channel gating.

2) Annotation: Of course, the presence of a lipidic moiety in the channel pore does not preclude its relevance. Previous ligand-gated ion-channel structures have also contained tubular pore densities, particularly in the presence of detergent. It is plausible that lipophilic agents in these structures occupy or even stabilize pre-existing cavities in functionally relevant states. Still, it seems inaccurate to ascribe to this open structure “similar pore geometry” (p. 5) as in previously reported homomeric receptors, or to claim that the “expanded open state pore geometry...shows that they are largely similar” (p. 6). The pore may be similar along its minimal dimension, but it is so expanded along its maximal dimension as to be contiguous with solvent. Structural lipids would presumably be required to stabilize this dissociated subunit interface, while avoiding the ion pathway (at least in the open state). Molecular simulations demonstrating stability and/or conductance of such structures could help to support their relevance, although—per the authors’ response to R1-C5 and R2-C3—configuration of such lipids might be complex. Similarly

dissociated transmembrane subunits in a bacterial-homolog structure were previously shown to be unstable in simulations, even with lipids inserted (PMID 36480474). In absence of computational evidence for stability and/or conduction, the structural annotation does not seem clearly justified.

It is recommended that the structures determined with glycine—currently annotated “open/expanded open/desensitized”—should be renamed e.g. “Gly-1/2/3,” with proposals as to their functional state(s) primarily limited to Discussion. If a speculative version of Fig. 6 is kept in the manuscript, it seems appropriate to include a structural lipid in the open state.

Response:

We employed the HOLE program¹⁻³ to calculate the inner dimensions along the full transmembrane pore, not only the minimal dimension at restriction points. We listed these minimal dimensions in Table S2 since these constriction points usually determine the functional states. The pore radii are indeed similar along the whole conduction pathways for respective functional states among heteromeric and homomeric channels (Sup. Fig. 6 l, m). In our open state structure, the widened subunit interface is not wide enough for ions to go through (~1.2 Å maximum radius) and thus the pore is not likely contiguous with solvent (please note that the distances shown in Fig. 2c-l are between counterpart C α atoms from adjacent subunits, not solvent-accessible gap between two subunits). In the expanded open state, the interface does become wide enough (~2.9 Å radius), but it is most likely blocked by the non-protein density/structural lipids at the very same position (Please see response to comment 1 and Supplementary Fig. 7). In summary, all currently available data, ours or in the literature, support our functional annotation.

Molecular simulation/AI-based computational approaches have progressed tremendously and are becoming more and more powerful, especially in analyzing transients/dynamics that are not resolved in static structures, or making predictions where structural/interaction data is not available. However, when simulation does not recapitulate existing experimental outcomes, it usually means not all relevant factors have been correctly accounted for in simulation, instead of experimental observations being not probably. In the specific case (PMID 36480474), such inconsistency of course maybe related to experimental conditions that were not considered in simulation (DDM detergent that results in expanded-open state of homomeric GlyRs, crystallization chemistry/contacts etc.), none of which are present in our experiments. Alternatively, it might also reflect current limitations (time scale, handling of water/salt/lipid and their interactions, local minimums etc.) of simulation where starting point affect outcome (Fig. S11 in PMID 36480474). In addition, physiological structures being static might be a flawed belief itself – specific states of certain proteins might be intrinsically dynamic. Overall,

we totally agree that simulation is extremely useful tool in many aspects, but might provide limited use without extensive experimental validation in our case.

Systematic and robust functional states assignment of the Cys-loop receptor family is a complicated task and surely require more investigation. Many questions remain. For instances, is the open structure we presented here the only open conformation, or are there other conformations, especially considering multiple sub-conductance states^{4,5}? Is the “expanded-open” structure physiologically relevant, and if so, is it conductive? What are the roles of non-protein densities in the pore? In addition, cation-selective Cys-loop receptors have more hydrophilic pores, making functional assignment more complicated.

In summary, we believe our assignments of functional states are supported by data in this study, and those available in current literatures. In acknowledgement of the complicated nature of heteromeric GlyR gating, we take the more prudent approach suggested here and changed the names of these structures into Gly-1,2,3, with the annotate functional states in the literature to which the pore is most similar to listed in parenthesis. We revised the main text to highlight this and discussed the functional state assignment, together with the non-protein density in the pore in the discussion section, with illustration (Sug. Fig. 7) showing these densities. In line with being more prudent, we did not include the lipid in open/expanded open states since more data is clearly required to identify and understand their role.

3) Mutagenesis: As described in response to R1-C3 and R2-C4 and Fig. 5, it is helpful that the authors show modifications at the α/α interface decrease channel activity. Still, these results do not clearly support the novel open structure, nor differentiate it from homomeric models. Repulsion at R(19')—the only residue on the outward-facing half of $\alpha 1:M2$ carrying a formal charge—is meant to initiate a pivot motion that dissociates B/C subunits from one another. However, the charge distributed on the complementary face of $\alpha 1$ is relatively weak (Fig. 4j), unlikely to overcome the energetic cost of disrupting van der Waals interactions across $3/4$ of the subunit interface. Repulsion among R(19') residues was previously proposed to promote expansion of the outward-facing end of the homomeric glycine-receptor pore (Ref. 42)—but not to disrupt hydrophobic interfaces distal to the proposed repulsion, nor stimulate insertion of structural lipids. Subsequent structures of homomeric $\alpha 1$ GlyRs (Refs. 11, 12) indicated that R(19') forms state-dependent polar/charged interactions with M1 and M3, and/or with lipophilic modulators such as ivermectin, suggesting multiple mechanisms by which this residue might influence function. Given that neutralization at R(19') is an established loss-of-function mutation in homomeric glycine receptors, possibly to an even greater extent than in heteromers (Ref. 8), this

effect cannot substantiate a distinctive mechanism of heteromer gating.

The authors also show that crosslinking $\alpha 1$ subunits at the level of -1'/1'—whose C β distance increases from 4 to 8 Å at one interface in the proposed opening transition—suppresses function. However, given the propensity of crosslinking to select nonphysiological nonfunctional states, these results would be more convincing with appropriate controls. If gating indeed relies on displacement at α/α but not α/β or β/α interfaces (as indicated by the charge-repulsion model), combinations of the $\alpha:A(-1')C$ or $V(1')C$ substitutions with $\beta:V(1')C$ or $A(-1')C$ respectively should be functionally insensitive to oxidation. Alternatively, if asymmetric gating is indeed a distinctive feature of heteromeric receptors, the $\alpha:A(-1')C+V(1')C$ substitution in homomeric receptors should be oxidation-insensitive.

Given the above concerns, it seems important to comment that perturbations at 19' and -1'/1' are consistent with the present structures, but do not clearly prefer them over previously proposed mechanisms. If possible, control experiments mutating β subunits and/or homomeric receptors should be shown.

Response:

The energetics of larger conformational changes might be complicated to evaluate. It makes sense that the charge distributed on the complementary face of $\alpha 1$ is too weak to overcome the energetic cost of disrupting van der Waals interactions across $3/4$ of the subunit interface – that is why this interface is intact in both apo and desensitized structures, and only one interface is widened in the open/expanded open state. However, the channel works cooperatively as a whole to channel the binding of multiple glycine into conformational changes that drives TM open. The opening of α - α interface is likely analogous to yielding of material at its weaker points under stress.

We agree that R19 mutants alone does not substantiate the asymmetric opening mechanism. However, combined with the structure, as well as cross-linking experiments, such mechanism becomes quite likely. We performed additional control experiments with mutations on the beta subunit, which allowed the crosslinking of α - β and β - α subunits near the intracellular side (revised Fig. 5e, f). Clearly crosslinking these two interfaces does not inhibit channel function. We have revised the text to include this additional experiment.

4) Ligands: In response to R1-C1, the authors argue that “local resolutions around the

glycine binding pockets...are similar across all the structures with sidechains well resolved...densities for glycine differ clearly in the sites denoted as occupied and unoccupied." However, the densities shown in *author response image 1, Supplementary Fig. 3c–d*, and the supplied maps do not look so clear. At most of these interfaces, density built as ligand appears not as a discrete volume, but as an extension of neighboring sidechains, potentially interchangeable with alternative rotamers. It does not seem like a coincidence that ligand densities are even less apparent in the lower-resolution structures, and that they do not necessarily overlap built ligands. The point is well taken that occupancy of all possible glycine binding sites is not required for activation, and it is helpful that the authors recognize "other activated conformations with different patterns of glycine occupancy are very likely present." On the other hand, they acknowledge "there are also subtle differences in side-chain conformations, but we are not confident in interpretation due to the limited resolution" (response to R2-C2); the ligand does not appear any better defined than the surrounding protein. Rather than building ligands at a likely underdetermined subset of possible sites, it seems more reasonable to presume that all five sites are partially occupied in these classes, with sufficient uncertainty to preclude definitive building.

Even in the fully occupied open state, glycine densities and poses differ between interfaces, and from previous studies. As supported by the authors' own functional data, previous studies have identified electrostatic interactions between R65 and the glycine carboxylate, and a potential cation- π contact between the electronegative face of F207 and the glycine amine. However, in this structure the carboxylate lies >4 Å from the R65 guanidinium in three of five subunits; the glycine amine orients either towards the electropositive edge F207, or away from it entirely. Other glycine contacts appear to be implausibly close, including <2.5 -Å intermolecular heavy-atom distances at all interfaces; at the C/D interface, the amine lies just 1 Å from F159. To be sure, a lack of definitive ligand densities does not negate the value of these structures. However, a more conservative approach to assigning stoichiometry and poses seems in order, especially at resolutions approaching 4 Å.

Given that the present data do not clearly justify modeling of glycine in some sites versus others, nor physically reasonable modeling of the ligand, it is recommended to remove glycine from the deposited models, and to reserve description of its likely (partially occupied) location(s) for Discussion.

Response:

Although density maps support our modeling of glycine in the binding pockets, we agree it is good to be more conservative. We have removed all glycine molecules from the desensitized state structure which has the lowest overall resolution (4.1 Å), and only pointed to the pockets where possible densities were observed (revised figure 2j). We also revised the discussion section to highlight the possibility of partial occupancy. We believe modeling of glycine in other density maps is reasonable and does contain information in the specific pattern of binding, which could fuel future studies. Multiple possible binding patterns do not necessarily mean outcomes of specific observations being always simple average, in addition to it currently remains a reasonable assumption.

References

- 1 Smart, O. S., Goodfellow, J. M. & Wallace, B. A. The pore dimensions of gramicidin A. *Biophysical journal* **65**, 2455-2460, doi:10.1016/S0006-3495(93)81293-1 (1993).
- 2 Smart, O. S., Neduelil, J. G., Wang, X., Wallace, B. A. & Sansom, M. S. HOLE: a program for the analysis of the pore dimensions of ion channel structural models. *J Mol Graph* **14**, 354-360, 376, doi:10.1016/s0263-7855(97)00009-x (1996).
- 3 Smart, O. S., Breed, J., Smith, G. R. & Sansom, M. S. A novel method for structure-based prediction of ion channel conductance properties. *Biophysical journal* **72**, 1109-1126, doi:10.1016/S0006-3495(97)78760-5 (1997).
- 4 Takahashi, T., Momiyama, A., Hirai, K., Hishinuma, F. & Akagi, H. Functional correlation of fetal and adult forms of glycine receptors with developmental changes in inhibitory synaptic receptor channels. *Neuron* **9**, 1155-1161, doi:10.1016/0896-6273(92)90073-m (1992).
- 5 Bormann, J., Rundstrom, N., Betz, H. & Langosch, D. Residues within transmembrane segment M2 determine chloride conductance of glycine receptor homo- and hetero-oligomers. *The EMBO journal* **12**, 3729-3737 (1993).

REVIEWERS' COMMENTS

Reviewer #2 (Remarks to the Author):

The addition of Supplementary Figure 7, control experiments in Figure 5, and related results/discussion text help to contextualize mechanistic proposals in this revised work. The modification of ligands and nomenclature for deposited structures also usefully acknowledge limitations to the interpretation of these data.

It remains difficult to reconcile the intersubunit gaps and non-protein densities observed in the so-called open state with a characteristic conducting state in the native functional cycle; the authors' claim e.g. that "all currently available data, ours or in the literature, support our functional annotation" of this state still seems overstated.

However, there seems to be little room to engage the authors on this larger point; and indeed, wilder, more wonderful scientific notions have turned out to be true. Hopefully future experimental evidence, and the critical consideration of the larger scientific community, will serve to clarify all these issues. I look forward to seeing this work in the public literature.

REVIEWER COMMENTS

Reviewer #2 (Remarks to the Author):

The addition of Supplementary Figure 7, control experiments in Figure 5, and related results/discussion text help to contextualize mechanistic proposals in this revised work. The modification of ligands and nomenclature for deposited structures also usefully acknowledge limitations to the interpretation of these data.

It remains difficult to reconcile the intersubunit gaps and non-protein densities observed in the so-called open state with a characteristic conducting state in the native functional cycle; the authors' claim e.g. that "all currently available data, ours or in the literature, support our functional annotation" of this state still seems overstated.

However, there seems to be little room to engage the authors on this larger point; and indeed, wilder, more wonderful scientific notions have turned out to be true. Hopefully future experimental evidence, and the critical consideration of the larger scientific community, will serve to clarify all these issues. I look forward to seeing this work in the public literature.

Response:

We acknowledge the widening of subunit interface, as well as asymmetric expansion of conduction pore are quite different from classic understandings. However, since activated heteromeric glycine receptor structural information was not available until very recently (which exhibit asymmetry to different extents), these features more likely reflect previously unobserved mechanism. Such mechanism is consistent with the chemical properties of different subunit

types, as well as the crosslinking experiments specifically directed at testing widening. As current data, ours and in the literature, can be explained with such mechanism, we believe it is a good model for understanding how heteromeric Cys-loop receptors work at present. We fully agree that much more research is required to test whether our observations represent a new general mechanism. The exact roles of non-protein materials in the function of this, and other ion channels, call for extensive further investigation.